# Water-mediated ion transport in an anion exchange membrane

Zhongyang Wang [1,2,10], Ge Sun[1,3,4,5,10], Nicholas H. C. Lewis [6], Mrinmay Mandal [7], Abhishek Sharma [1,8], Mincheol Kim [1], Joan M. Montes de Oca [1], Kai Wang[1], Aaron Taggart[9], Alex B. Martinson [9], Paul A. Kohl[7], Andrei Tokmakoff [6] ✉, Shrayesh N. Patel [1] ✉, Paul F. Nealey [1,9] ✉ & Juan J. de Pablo [1,3,4,5,9] ✉

Water is a critical component in polyelectrolyte anion exchange membranes (AEMs). It plays a central role in ion transport in electrochemical systems. Gaining a better understanding of molecular transport and conductivity in AEMs has been challenged by the lack of a general methodology capable of capturing and connecting water dynamics, water structure, and ionic transport over time and length scales ranging from those associated with individual bond vibrations and molecular reorientations to those pertaining to macroscopic AEM performance. In this work, we use two-dimensional infrared spectroscopy and semiclassical simulations to examine how water molecules are arranged into successive solvation shells, and we explain how that structure influences the dynamics of bromide ion transport processes in polynorbornene-based materials. We find that the transition to the faster transport mechanism occurs when the reorientation of water molecules in the second solvation shell is fast, allowing a robust hydrogen bond network to form. Our findings provide molecular-level insights into AEMs with inherent transport of halide ions, and help pave the way towards a comprehensive understanding of hydroxide ion transport in AEMs.

Anion exchange membranes (AEMs) are at the heart of electrochemical conversion and storage devices such as fuel cells[1], water electrolyzers[2], $CO_2$ electrolyzers[3], redox flow batteries[4], and reverse electrodialysis[5]. An AEM is composed of a hydrophobic polymer backbone and hydrophilic cationic groups that allow anions to permeate while rejecting cations. In an AEM, the transport of anions is facilitated by water molecules. Efficient operation of electrochemical systems having AEMs as separators and electrode materials therefore requires a thorough understanding of water absorption and ion transport under different levels of hydration.

Recent studies on AEMs, including both experimental and modeling approaches, have significantly advanced our understanding of water absorption and ion transport mechanisms[6–8]. It is believed that water absorption is governed by the Park model[9], in which Langmuir absorption occurs at low relative humidity (RH), Henry-type absorption occurs at moderate RH, and clustering absorption takes over at

[1]Pritzker School of Molecular Engineering, University of Chicago, Chicago, IL, USA. [2]Department of Chemical and Biological Engineering, The University of Alabama, Tuscaloosa, AL, USA. [3]Department of Chemical and Biomolecular Engineering, Tandon School of Engineering, New York University, Brooklyn, NY, USA. [4]Department of Computer Science, Courant Institute of Mathematical Sciences, New York University, New York, NY, USA. [5]Department of Physics, New York University, New York, NY, USA. [6]Department of Chemistry, James Franck Institute, The University of Chicago, Chicago, IL, USA. [7]School of Chemical and Biomolecular Engineering, Georgia Institute of Technology, Atlanta, GA, USA. [8]Department of Chemical Engineering, The Cooper Union for the Advancement of Science and Art, New York, NY, USA. [9]Materials Science Division, Argonne National Laboratory, Lemont, IL, USA. [10]These authors contributed equally: Zhongyang Wang, Ge Sun. ✉e-mail: tokmakoff@uchicago.edu; shrayesh@uchicago.edu; nealey@uchicago.edu; jjd8110@nyu.edu

high RH. In Langmuir absorption, water molecules occupy vacancies in the anion solvation shell, and start screening the anion-cation interactions. In this regime, the water state is conceptualized as "bound water" or, alternatively, it is described as "highly polarized water". Clustering absorption occurs when water molecules start forming hydrogen bonded clusters, after all vacancies on the solvation shell are occupied. In this regime, the hydrogen bonded network is presumed to be similar to that of bulk water[10]. Depending on the water absorption level, anion transport in an AEM is governed by three transport mechanisms: surface site hopping, vehicular transport, and Grotthuss hopping (OH⁻ transport)[3,8,11,12]. At low hydration levels, the hopping of anions between solvation sites that consist of cationic groups (e.g., $N^+$, $P^+$, and $S^+$) is dominated by the surface site hopping mechanism[13]. The surface site hopping mechanism is correlated with the segmental mobility and solvation environment of the polymer chain[14–16]. When the polymers are further exposed to humid environments, the vehicular and Grotthuss mechanisms begin to govern the overall anion transport. The vehicular mechanism involves concentration gradient-driven diffusion and electromigration, both dependent on the diffusion coefficient of ions moving through the membrane. In addition to vehicular transport, OH⁻ transport is facilitated by the Grotthuss (proton hopping) mechanism[11,17]. Water plays a crucial and complex role in these processes, including solvating anions, plasticizing polymers, forming hydrogen bonds (H-bonds) with anions, and clustering with other water molecules to form bulk-like configurations that enable vehicular diffusion and Grotthuss hopping (for OH⁻ transport only)[3]. Overall, the discussion above serves to underscore that ion transport and water structure are tightly coupled in AEMs, and it is therefore critical to develop a detailed understanding of the molecular-level processes at play.

In this work, we use a combination of computational and experimental techniques to study hydration-dependent bromide ion transport and water dynamics in thin films of polynorbornene-based AEMs (Fig. 1a)[6,9]. As explained above, a key aspect shared by both the Grotthuss and vehicular mechanisms is the integral role of water dynamics, water structure, and ion solvation in ion transport. By focusing on bromide ion transport, we intentionally excluded the Grotthuss mechanism to thoroughly investigate these factors without the influence of proton hopping. We believe that a clear understanding of these effects is important before directly probing OH⁻ transport. Water uptake and AEM thin-film expansion are measured as a function of RH through in situ ellipsometry. The resulting changes in ionic conductivity are measured by electrochemical impedance spectroscopy (EIS)[18]. Note that EIS captures ion dynamics on timescales ranging from microseconds to seconds. Ultra-fast two-dimensional infrared (2D IR) spectroscopy is used to measure the reorientation and fluctuations of water molecules on femtosecond-to-picosecond timescales[19–22]. We find that ion transport can be categorized into a slow regime and a fast regime. We examine the molecular contributors to the transition from slow to fast dynamics in terms of ion solvation, water dynamics, and water percolation networks by relying on molecular dynamics (MD) simulations, coupled to a semiclassical model for molecular vibrations that provides a quantitative interpretation of spectroscopic signals. Here we note that quasi-elastic neutron scattering[23] has previously been used to examine water dynamics at picosecond timescales, and broadband electrical spectroscopy has been employed to investigate ion conducting pathways through grain boundaries in polyelectrolytes[24–26]. A new advance introduced here for the study of ionic transport in electrochemistry is the synergistic use of 2D IR and semiclassical simulations to expand the current knowledge of water

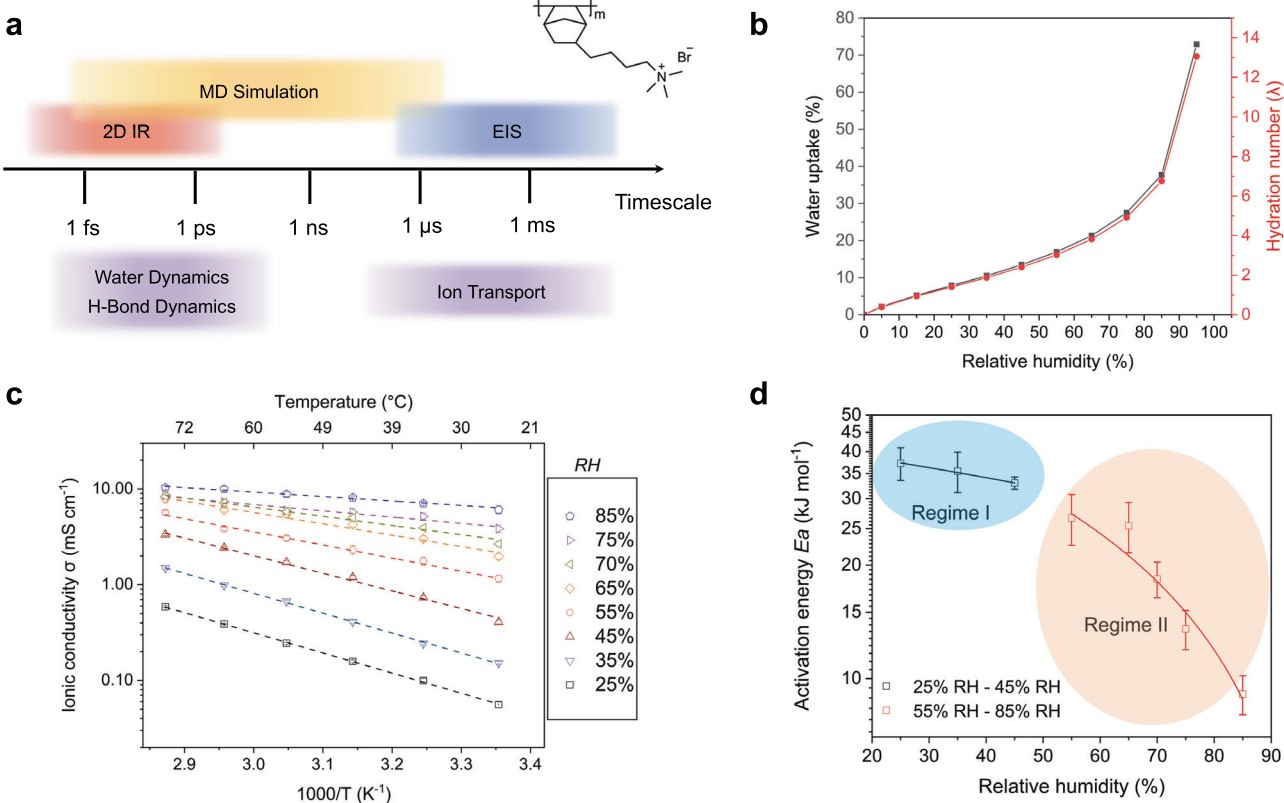

**Fig. 1 | Ion conductivities of the AEM. a** Schematic diagrams of simulation and experimental methods used to probe dynamics of H-bonding, water molecule, and ion transport at different time scales. The chemical structure of PBBNB⁺Br⁻ is shown in the figure. The chemical structure of the AEM, PBBNB⁺Br⁻, is shown on the upper right. **b** Water uptake and hydration number as a function of relative humidity (RH). **c** Ionic conductivities of PBBNB⁺Br⁻ as a function of temperature and RH. Error bars represent standard deviations obtained with three samples. **d** Change of activation energy ($E_a$) as a function of RH.

dynamics and ion transport mechanisms from femtoseconds to picoseconds to microseconds within state-of-the-art AEMs.

## Results

### Ion transport in anion exchange thin films

Poly(bromo butylnorbornene) (PBBNB) was polymerized from bromobutyl norbornene (BBNB) via vinylic addition, and spin-coated on top of interdigitated electrode arrays (IDEs). After a vapor infiltration reaction (VIP), PBBNB was converted to PBBNB$^+$Br$^-$ by a quaternization reaction with trimethylamine. Material synthesis and characterization details are provided in Supplementary Section 1–3. The water uptake of the AEMs as a function of RH is shown in Fig. 1b. The exponential increase in water uptake after 65% RH is generally attributed to water clustering. It is commonly believed that the formation of free water, also referred to as unbound water, is a prerequisite for the vehicular transport mechanism[3]. However, it remains unclear whether the formation of water clusters directly indicates the presence of free water. In the following sections, we examine in more detail the water structure and examine the underlying transport mechanisms in the context of our MD and 2D IR results.

To obtain the film's ionic conductivity $\sigma$, EIS measurements were performed on the IDE sample over a temperature range of 25 °C to 75 °C and controlled RH levels from 25% to 85% (Fig. 1c). It is important to note that PBBNB$^+$Br$^-$ remains in a glass state throughout all EIS measurements, as no glass transition temperature ($T_g$) was detected in the differential scanning calorimetry (DSC) measurements (Supplementary Fig. 20 and 21). In addition, we assume our polymer system to be homogeneous, as there is no evidence of crystallinity, phase separation, or ordered/disordered domains (Supplementary Fig. 24). Therefore, the plot of ln $\sigma$ versus $1000/T$ follows Arrhenius behavior for each RH, allowing us to fit the data and determine the activation energy ($E_a$):

$$\sigma = \sigma_e \, e^{\frac{-E_a}{RT}} \tag{1}$$

where $\sigma_e$ is the pre-exponential factor, $R$ is the gas constant, and $T$ is the absolute temperature.

Figure 1d shows the calculated $E_a$ as a function of RH. At 25% RH, $E_a$ is 37.2 kJ mol$^{-1}$ and decreases slowly to 33.0 kJ mol$^{-1}$ at 45% RH. From 55% RH to 85% RH, by further increasing the water content, $E_a$ decreases rapidly from 26.7 kJ mol$^{-1}$ to 9.1 kJ mol$^{-1}$. The onset of a rapid decrease of $E_a$ with increasing humidity in the high hydration regime suggests that the observed trends in $E_a$ can be classified into regimes I and II, as highlighted in Fig. 1d. To interpret our measurements, we adopt a classical percolation theory and propose a scenario where a threshold volume fraction of aqueous phase exists at *ca.* 25% RH in the polymer, below which ion flow is impeded[27]. To describe the transition between different ion transport mechanisms, we turn our attention to the water structure in the next section.

### Understand local ion solvation environments

The results of simulations for volumetric expansion and ionic conductivity as a function of RH can be compared to our experimental measurements, as shown in Supplementary Fig. 26. There is good agreement between simulations and experiments, serving to validate our choice of forcefield. Figure 2a–c shows the radial distribution functions ($g(r)$), from which we can extract the radii of the first solvation shells of Br$^-$, and the associated coordination number ($CN$). The radius of the first solvation shell between Br$^-$ and water oxygens is 4 Å, and that between Br$^-$ and nitrogen in quaternary ammonium groups is 6.5 Å, both of which remain unchanged as a function of RH.

Figure 2d–f visually translate the binding motifs of Br$^-$ through heatmaps. These patterns arise from the coordination of quaternary ammonium groups and water molecules in the first solvation shell of Br$^-$.

As RH increases, a transition emerges: the presence or relevance of the ammonium groups diminishes as the number of water molecules increases, a trend that can be visualized from the bottom right to the upper left on the heatmap. This transition, illustrated through representative snapshots in Fig. 2g that depict the most probable binding motifs corresponding to each RH level, suggests that water molecules play an important role in the solvation of anions by reducing their interaction with ammonium groups. Notwithstanding these changes, the Br$^-$ ions are never fully dissociated from the cationic groups and water molecules reside within either the first or second solvation shell across all systems shown in Supplementary Fig. 29, which goes against the existence of bulk-like water in our materials, even at the highest RH level.

We now turn to Fourier-transform infrared (FTIR) spectroscopy to address several of the insights provided by our simulations. In Fig. 2h we present the absorption spectrum of 10% HOD in H$_2$O for PBBNB$^+$Br$^-$. The OD stretching mode of dilute HOD in H$_2$O serves as a local vibrational probe of the environment, including the hydrogen bonding network. It also provides a measure of the local electrostatic field imparted on the OD bond by the solvation environment[28–30]. As the RH increases, a subtle redshift in the peak frequency, together with an increase in the intensity on the red side of the spectrum, indicates an increase in the population of water-water hydrogen bonds[31,32]. Nevertheless, the spectrum maintains a narrow profile, with frequencies significantly higher than those observed in bulk water, signaling the disruption of the hydrogen bond network by the polymer and the Br$^-$, and negating the existence of a significant amount of free water across all RH levels. With the addition of water, no distinct characteristic peaks are observed corresponding to phase separation, even at 95% RH, as evidenced by Supplementary Fig. 24.

To interpret the IR spectra discussed above, we turn to mixed quantum-classical approach that is capable of mapping the vibrational frequency to the local electric field experienced by the O-D vibrational coordinate and yields the absorption spectrum from the dipole correlation function. The simulated IR spectra, depicted in Fig. 2f, are in good agreement with the experimental measurements in Fig. 2e, serving to validate the structures predicted in simulations and reinforcing the absence of bulk-like water in the system. This is further supported by the DSC results in Supplementary Fig. 21.

### Water percolation analysis by graph theory

In this section we rely on a graph theoretic approach to elucidate the ways in which local hydration environments shape the global percolation network, which is visualized in Fig. 3a, b. More specifically, we use the $k$-stub[33] method to quantify percolation. In this framework, bromide ions and water molecules are treated as nodes in our ion-water network, interconnected by edges represented by hydrogen bonds. The parameter $k$ represents the minimum number of connecting edges between nodes. We then determine the percolation "robustness" by calculating the percolation probability at varying $k$ values ($P_k(x)$), defined as the ratio of the largest connected component ($LCC$) to the total component, as detailed in Supplementary Fig. 31a. Note that, when $k = 1$, the calculation gives the trivial percolation probability. We systematically remove nodes with degrees less than a given $k$ value, and we update iteratively the $LCC$ and percolation probability $P_k(x)$ throughout the process. This procedure is visualized in Fig. 3c–f, which provides the cross-sections of the simulated system, each corresponding to different degrees of connectivity. Figure 3c, d present snapshots of the system at $k = 1$, illustrating the initial network graph at two distinct RH levels: 45% and 85% corresponding to Regime I and II. As we increase $k$ to 3, as shown in Fig. 3e, f, the impact of excising nodes with two or fewer edges becomes evident. At 45% RH, this leads to a significant fragmentation of the large network, consequently reducing the $LCC$ and approaching a zero-percolation probability. Conversely, the network at 85% RH demonstrates resilience by

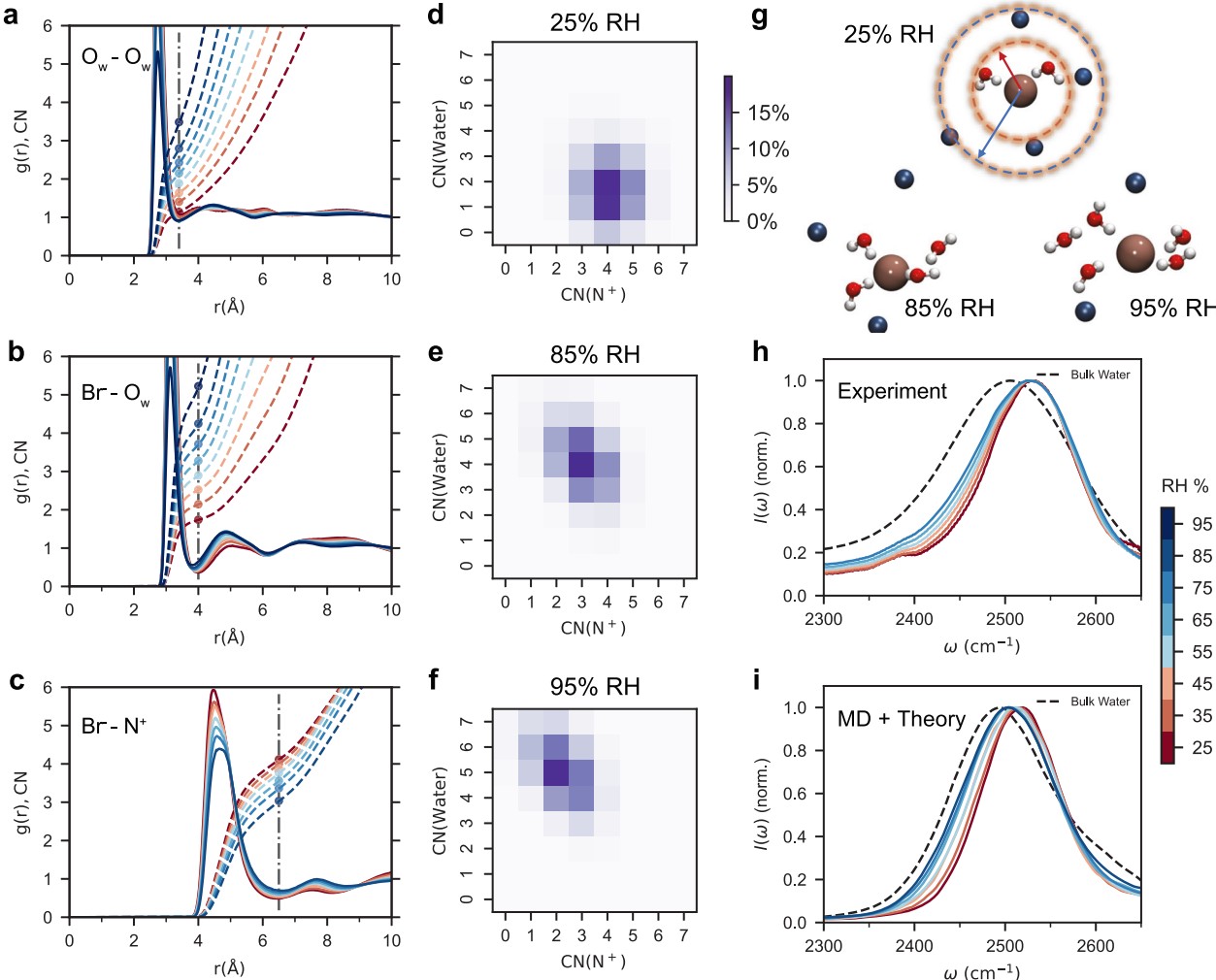

**Fig. 2 | Solvation environments of bromide ions and molecular interactions in PBBNB⁺Br⁻.** Radial distribution functions ($g(r)$, solid curves) and coordination numbers (*CN*, dashed curves) for interactions of **a** $O_w$-$O_w$, **b** $Br^-$-$O_w$, and **c** $Br^-$-$N^+$. Dashed vertical lines and intersecting dots represent the first solvation shell radius and its coordination number, respectively. **d**–**f**, Heatmaps of coordination numbers for water and N⁺ across RH levels from 25% to 85%. The x and y-axes represent the coordination numbers of the ammonium groups and water molecules,

respectively, while color intensity within each grid indicates the probability of observing each binding motif. **g** Representative most probable binding motifs at 25%, 85% and 95 % RH. The brown and blue spheres represent Br⁻ ions and nitrogen atoms, respectively. The red and blue dashed circles illustrate the first solvation shell radii involving water oxygen and quaternary ammonium nitrogen, respectively. **h** Experimental and **i** MD simulated IR absorption spectra of 10% HOD in H₂O in the OD stretch region.

maintaining a significant percolation probability and a robust *LCC*. Supplementary Fig. 32 clarifies this analysis even further through a detailed stepwise progression and incorporating additional comparative cases.

The transition from $k = 1$ to 3, illustrated in Fig. 3g, marks a critical point in our analysis. Systems with lower RH (25–45%) exhibit a sharp decline in $P_k(x)$ while those above 55% maintained high levels. This trend, corroborated by our experimental data, reveals several key characteristics of these systems. Specifically, the networks at lower RH environments are vulnerable to disintegration as $k$ rises to 3. In contrast, the networks for higher RH remain robust, consisting primarily of nodes with at least three edges. These findings highlight the central role of 3-edge nodes in ion transport.

While Fig. 3a–f show the structural characteristics of the ion-water network, we now turn our attention to ion dynamics. Figure 3h illustrates the changes in ion mobility across two distinct RH regimes. It presents the distribution of the first passage time (FPT), which is the time required for an ion to traverse 1 nm, at various RH levels. Notably, between 25 and 45% RH, the distribution curves overlap each other, with no significant discernible peaks. In contrast, for 55–85% RH, the

FPT decreases, and the peaks become more pronounced and narrower, indicating the marked increase in ion mobility associated with rising RH levels. Figure 3i further reveals that at 85% RH ions travel greater distances compared to those at 45% RH. At a lower RH, a significant portion of the ion displacement can be attributed to hopping events[13] occurring on time scales of approximately 15 ns. In comparison, a higher RH promotes continuous diffusion through less restrictive pathways, resulting in larger ion displacements. Supplementary Fig. 36 shows the nuanced differences in ion solvation structures and dynamics of the two mechanisms. These findings, grounded in both experimental and simulation results, encapsulate the microstructural shifts in percolation robustness that drive changes in conductivity.

**Water dynamics analysis by 2D IR**

We have demonstrated that the Br⁻ transport takes place in an environment that is highly confined by the polymer and does not resemble bulk-water. Water dynamics are closely linked to the ions motion. We have used 2D IR spectroscopy to investigate how the local solvation environment alters the electrostatic field, and thus affects water

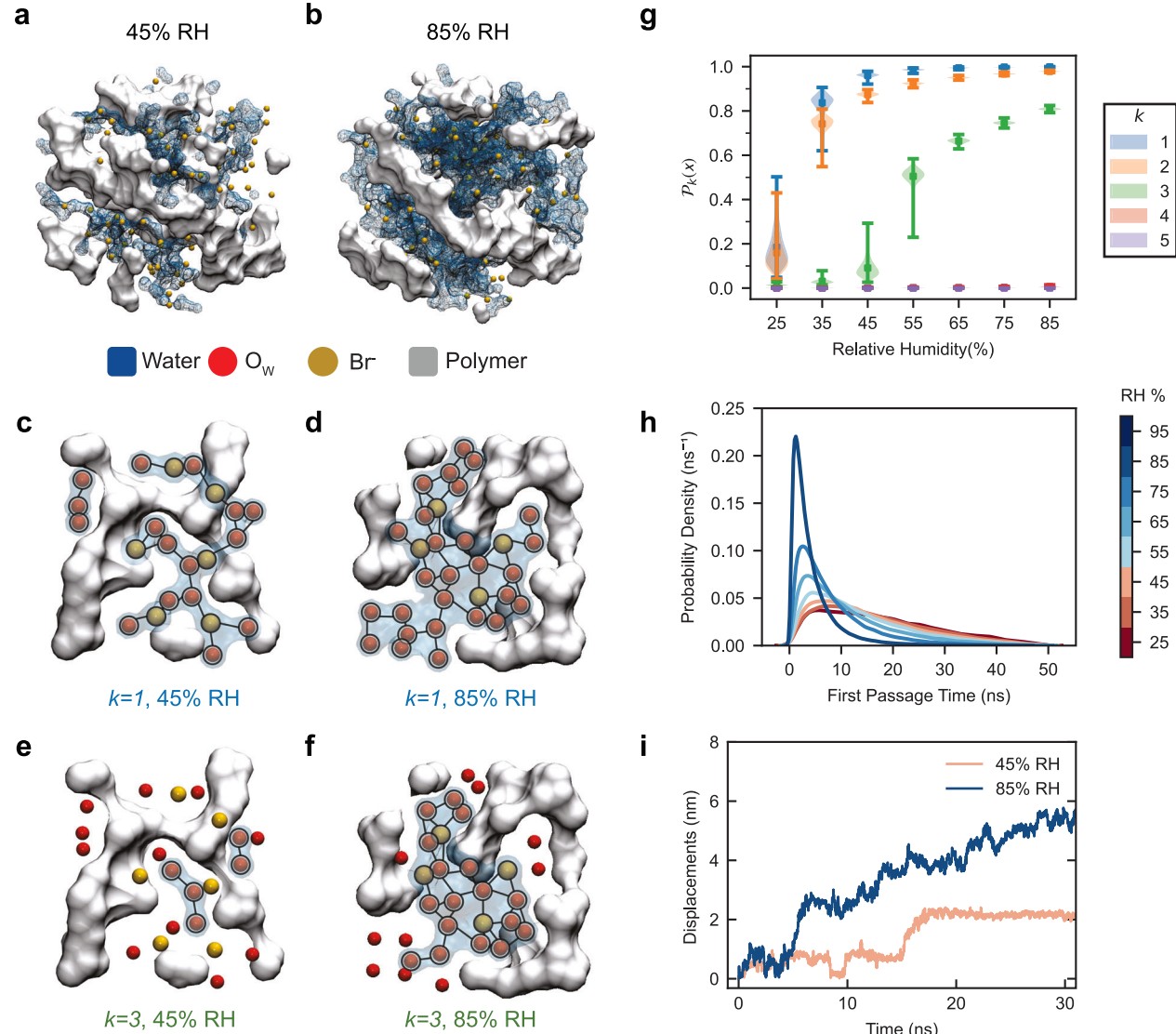

**Fig. 3 | Robustness of percolation in the ion-water network analyzed by graph theory.** MD simulation snapshots of the system at **a** 45% and **b** 85% RH, highlighting the visualization of the increased water percolation at higher RH. *k-stub* analysis snapshots at *k* = 1 for the polymer system **c** at 45% and **d** 85% RH, and at *k* = 3 for **e** 45% and **f** 85% RH. Nodes (black circle, $O_w$ or $Br^-$) and edges (black lines, H-bonds) illustrate the ion-water network, which disintegrates into smaller groups at 45% RH at *k* = 3 due to bottleneck presence, while the 85% RH system maintains a robust and percolated network. **g** Percolation probabilities at various *k* values. When *k* = 3, $P_3(x)$ is almost 0 below 45% RH and starts to increase above 55% RH. Error bars represent the standard deviation computed from simulations sampled over 1000 frames. **h** First Passage Time (FPT) distribution for ion mobility across RH levels in a simulated environment from 25% to 95%. Each distribution details the time to cross a 1 nm distance. **i** Displacement [$r(t) - r(0)$] trajectories of representative ions over a 30 ns simulation period at 45% and 85% RH.

motions. Figure 4a presents selected 2D IR spectra corresponding to the OD stretching mode in HOD. Note that we also simulated the spectra for the same system at the same waiting time $t_2$, as shown in Fig. 4b. These spectra show how the slope of the center line decays with increasing $t_2$. The centerline slope (CLS) captures spectral diffusion changes caused by fluctuating electrostatic fields[34–36], while the Frequency-Frequency Correlation Function (FFCF) provides a statistical measure of how these frequency changes correlate over time due to the dynamic fluctuations in the local environment[37]. In Fig. 4c, d, we present the RH dependent CLS decay kinetics extracted from our experiments and the FFCF determined from MD simulations, respectively. Both show similar trends as a function of RH, with the relaxation accelerating as the water content in the polymer is increased. Moreover, FFCF results from our simulations exhibit two distinct regimes, with a noticeable gap between low and high RH. Because the motions of the polymer and the $Br^-$ ions are much slower than the decays we observe here in the CLS, the relaxation is driven primarily by the

movement of the water molecules[38]. These findings help explain how the electrostatic field and hydration level affect the dynamic behavior of the system.

We also study the motions of water molecules using polarization dependent IR transient absorption (TA), which probes the hydrogen bond switching dynamics in the polymer[39–41]. Figure 4e, f show both experimental and simulated results for the TA anisotropy decay across all RH levels, reflecting the orientational dynamics of the probed water molecules. These results reveal a distinct division between low and high RH regimes, which is also seen in our simulations. Within the 25–55% RH range, we observe a tight grouping of all curves, indicating similar time scale relations and amplitudes and implying that lower RH conditions foster slow water orientational dynamics; we attribute this behavior to the constraints enforced by the presence of two waters (two edges) forming a bottleneck, as evidenced in Fig. 3. In higher RH environments, above 55%, we find fewer bottlenecks and faster dynamics.

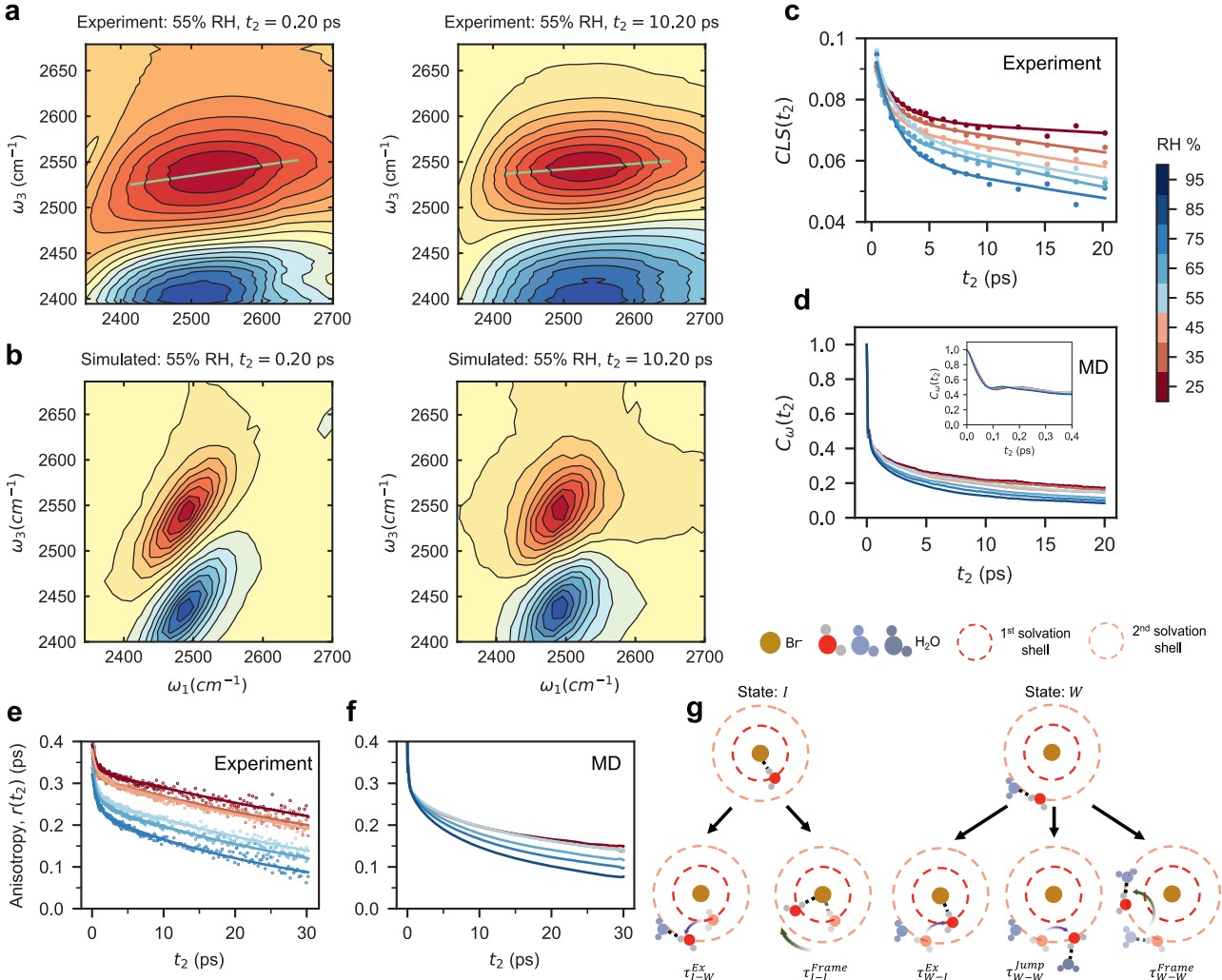

**Fig. 4 | Local electrostatic field determines water molecule reorientation motions in PBBNB⁺Br⁻. a, b** Selected 2D IR spectra at 55% RH at waiting times $t_2 = 0.20$ and 10.20 ps from experiments and simulations, respectively. The 0–1 transitions are shown with yellow-red contours, and the 1–2 transitions are shown with blue contours. The light green lines indicate the center lines used to extract the CLS decays. Additional 2D IR spectra can be found in Supplementary Fig. 35. **c** Center Line Slope (CLS) extracted from 2D IR spectra. **d** Frequency-frequency correlation function (FFCF) simulated by MD. The inset presents an enlarged view of the FFCF curves on a shorter timescale for clarity. **e, f** Anisotropy decay $r(t)$ from Ultra-fast IR and MD simulations, respectively. **g** Schematic illustration detailing reorientation motions from two initial states: State $I$ (water molecule sits in the 1st

solvation shell and interacts with the bromide ion through H-bond) and State $W$ (water molecule sits in the 2nd shell and interacts with the adjacent water molecule through H-bond). One type of water reorientation includes jumps through breakage and formation of H-bonds, as indicated by a purple arrow. The associated relaxation times are $\tau^{Ex}$ (water molecule jumps from one solvation shell to another) and $\tau^{Jump}$ (water molecule jump within the same solvation shell). The other type of water reorientation includes co-rotations of a water molecule with intact H-bonds, as indicated by a green arrow. The associated relaxation time is $\tau^{Frame}$. The black dashed line corresponds to H-bond, and light blue and gray spheres correspond to initial and new H-bond acceptors, respectively.

Figure 4e, f illustrate the collective effects of water motion on the anisotropy decay observed for all water molecules. To understand the underlying causes behind these two distinct regimes, we can decouple the rotational motions of water molecules[42]. Since water molecules reside in either the first or second solvation shell of Br⁻ ions in our AEM, we define two initial states for water OH-bonds, state $I$ and $W$, representing H-bonding to a bromide ion or another water molecule, respectively[43,44]. The transition dynamics from state $I$, depicted on the left of Fig. 4g, involves two reorientation processes. The first process involves a large-amplitude angular jump, denoted by $\tau^{Ex}_{I-W}$, prompting a H-bond shift from an ion to a water molecule and inducing a state alteration. The second process, represented by $\tau^{Frame}_{I-I}$, is a localized frame reorientation alongside the bromide ion but maintaining the existing state. Starting from state $W$, shown on the right of Fig. 4g, the reorientation involves three processes. One is the reverse of the process in state I, denoted by $\tau^{Ex}_{W-I}$. The other two, $\tau^{Jump}_{W-W}$ and $\tau^{Frame}_{W-W}$,

correspond to reorientations observed in liquid water and preserve the state $W$ in the second solvation shell.

By decoupling these modes, we find that water behaving as 'W-W Jump' and 'W-W Frame' exhibits this bi-modal characteristic within our observational timeframe; additional details are presented in Supplementary Fig. 37. In low RH environments, there are very few water molecules in the Br⁻ second solvation shell, and they primarily form H-bonds with the water molecules that are in the first shell. These second-shell water molecules function as bridges ("bridge" waters, in the 2nd shell) that link adjacent first-shell water molecules ("linked" waters, in the 1st shell), consequently restricting their mobility. Notably, the "bridge" waters exhibit a consistent reorientation timescale on the order of tens of picoseconds, as shown in Supplementary Fig. 37, and they govern the first regime observed in the anisotropy decay shown in Fig. 4e, f. This timescale aligns with the local fluctuations of Br⁻ ions shown in Fig. 3i. As time progresses from tens of picoseconds

to nanoseconds, these local interactions accumulate, allowing Br$^-$ ions to traverse nanometer-length scales, as displayed in Fig. 3h, i. In other words, ion transport over nanometers in nanoseconds is the cumulative result of these shorter timescale events involving changes in the local solvation environment.

Since Br$^-$ ions are solvated by the "linked" waters in the first shell, their dynamics are slowed due to the low mobility of these "linked" waters, leading to a higher energy barrier that must be overcome for long-distance transport. This higher energy barrier corresponds to the calculated $E_a$ in Regime I. Conversely, at higher RH levels, as the water population in the second shell increases, the bridging effect is reduced, resulting in faster orientational dynamics for the first-shell waters. This accelerates local ion fluctuations and reduces the energy barrier for ion transport, transitioning into Regime II. This molecular-level description demonstrates how events at different timescales are interconnected, and manifests the central role of the 3-edge ($k = 3$) in our $k$-stub theoretical analysis, connecting it to the observed dynamics.

## Discussion

The ionic conductivity of PBBNB$^+$Br$^-$ exhibit Arrhenius behavior; we have determined the corresponding activation energy $E_a$ as a function of RH. We have found that the change of $E_a$ as a function of RH can be classified into two regimes: a slower ion transport Regime I (25% RH ~ 45% RH) and a faster ion transport (>55% RH) Regime II. By combining results from $g(r)$, percolation theory, graph theory, and ultrafast 2DIR, we have arrived at a comprehensive picture of the solvation structure in these materials, and the dynamics of water and ions. To the best of our knowledge, this study represents a first use of 2DIR in the context of ion transport in hydrated polymers, and our results show that it is a powerful tool for characterization of these types of materials. In this particular study, we have been able to determine that the slower transport mechanism in Regime I corresponds to site hopping. In this state, a water percolation network does not exist, and the population of water molecules in the 2nd solvation shell is minimal. The Br$^-$ ions are trapped in local solvation sites, and ion transport is controlled by conformational fluctuations of the polymer side chains. Contrary to the current general understanding, the faster ion transport in Regime II does not correspond to a vehicular transport mechanism that requires the presence of bulk water or free water. We find that the faster ion transport mechanism is enabled by the formation of a robust water network with at least 3 edges ($k = 3$), and a large population of water molecules in the 2nd solvation shell that reduce the bridging effect and the constraints on the ions. Our findings reveal that water molecules in the 2nd solvation shell exhibit increased mobility, enabling them to transition across different solvation sites and enhancing ion mobility.

The Br$^-$ ion transport mechanism has important implications for processes such as reverse electrodialysis[45] and redox flow batteries[46]. Our results indicate that, to enable fast ionic transport even at low RH, the design of next-generation AEMs should focus on increasing the local (not global) concentration of water molecules in the AEM, thereby enhancing water dynamics and the robustness of the water percolation network. This work provides molecular insights into OH$^-$ transport in an AEM, where the local water concentration of a block copolymer-based AEM is higher than that in the analogous homopolymer; the global water concentration in the homopolymer, however, is higher than that in the block copolymer[47]. For study of OH$^-$ transport, other computational methods, particularly quantum mechanical approaches such as ab initio MD and quantum mechanics/molecular mechanics (QM/MM), or alternative approaches based on machine learning (ML) potentials, are necessary to capture proton hopping mechanisms. By combining these quantum mechanical and ML methods with classical MD simulations, we hope to achieve in future work a comprehensive understanding of OH$^-$ conducting AEMs.

## Methods

### Synthesis and preparation of polynorbornene-based anion exchange thin films

The poly(bromo butylnorbornene) homopolymer (PBBNB) was synthesized from bromobutyl norbornene (BBNB) in a nitrogen filled glove box. In a mixture of toluene (0.5 g, from Sigma-Aldrich, ACS reagent, ≥99.5%) and trifuorotoluene (TFT) (0.5 g from Sigma-Aldrich, 99%), (η3-allyl)Pd(iPr3P)Cl (12 mg, 0.034 mmol from Sigma-Aldrich) and lithium tetrakis(penta-fuorophenyl)-borate·(2.5Et2O) (Li[FABA]) (28 mg, 0.031 mmol from Boulder Scientific Co.) were mixed in 1:1 mole ratio to make the catalyst solution. The mixture was stirred for 20 min to generate the cationic Pd catalyst for initiating the polymerization. The monomer, BBNB (0.45 g) was purified through three freeze–pump–thaw cycles. Next, toluene (10 mL) was added to make a 5 wt% solution of the monomer. The catalyst solution was injected at once into the BBNB solution under vigorous stirring. The solid polymer was collected after precipitating three timesin methanol (from Sigma-Aldrich, ACS reagent, ≥99.8%). The polymer product was dried under vacuum at 60 °C and the obtained polymer has a molecular weight ($M_n$) of 68 kg mol$^{-1}$ with dispersity (Đ) of 1.20, as determined using a polystyrene standard. The size exclusion chromatography (SEC) curve is shown in Supplementary Fig. 3 and the $^1$H-NMR spectra of the obtained PBBNB is shown in Supplementary Fig. 2.

All substrates used in this study were cleaned by ultrasonication in acetone (from Sigma-Aldrich, ACS reagent, ≥99.5%) and 2-propanol (from Sigma-Aldrich, ACS reagent, ≥99.5%) for 5 min in each solvent. To obtain thin film samples, neutral PBBNB were first dissolved in chlorobenzene and spin-coated on different substrates. For example, PBBNB was spin-coated on Si substrates with 1.5 nm SiO$_2$ for ellipsometry measurements, on interdigitated electrode arrays (IDEs) for EIS measurements, and on Au coated Si substrates for FTIR measurements. Then, all substrates were annealed at 80 °C to remove residual solvent. The thickness of neutral PBBNB film is *ca.* 65 nm, as confirmed by the ellipsometry at ambient environment. After quaternization reaction, PBBNB$^+$Br$^-$ anion exchange thin film has a thickness of *ca.* 98 nm, as confirmed by the ellipsometry at ambient environment.

### Size exclusion chromatography

Size exclusion chromatography (SEC) experiments were performed on a Wyatt/Shimadzu SEC. THF (from Sigma-Aldrich, ACS reagent, ≥99%) was used as eluent at a flow rate of 1 mL min$^{-1}$. The differential refractive index signal was collected with Wyatt Optilab T-rEX differential refractive index detector and calibrated against a polystyrene standard. The SEC curve is shown in Supplementary Fig. 3.

### Nuclear magnetic resonance

Nuclear magnetic resonance (NMR) measurements were carried out on a 400 MHz Bruker Avance-III-HD nanobay spectrometer equipped with a BBFO SmartProbe and 24-sample SampleCase autosampler, using Topspin 3.6.2. The PBBNB sample was prepared by dissolving approximately 30 mg of polymer in 1 mL of deuterated chloroform (from Sigma-Aldrich). Tetramethylsilane (TMS) (from Sigma-Aldrich, ACS reagent, NMR grade ≥99.9%) was used as the internal standard for calibrating the chemical shift ($\delta = 0$ ppm for $^1$H). The $^1$H-NMR spectra are shown in Supplementary Fig. 2.

### In situ spectroscopic ellipsometry

J.A. Woollam alpha-SE spectroscopic ellipsometer equipped with a liquid cell and relative humidity generator was used to measure the thicknesses of PNB-based anion exchange films. The humidity generator (RH95) was purchased from Linkam company. Supplementary

Fig. 11 shows the experimental setup used to measure film thickness at different hydration levels. A Cauchy layer model was used to derive the thicknesses and optical properties of polymer thin films. The experimental result is shown in Supplementary Fig. 12.

## Water uptake

Water uptake measurements were performed by a dynamic vapor sorption equipment (Intrinsic) provided by Surface Measurements Systems company. The temperature range of the instrument is from 20 °C to 40 °C. The RH range of the instrument is from 0% RH to 98% RH. The results are shown in Supplementary Figs. 8–10.

## Electrochemical Impedance Spectroscopy (EIS)

EIS measurements were performed on top of IDEs with $1\,\mu m$ $SiO_2$ supporting layer using Gamry 600+ potentiostat inside a humidity chamber (ESPEC SH-242). The fabrication details can be found in our previous report[27]. The EIS was measured from 1 MHz to 0.1 Hz at different temperatures. The ionic resistance ($R_{ion}$) data were then extracted from the impedance spectrum by fitting an equivalent circuit shown in the Supplementary Discussion section. The ionic conductivity of the thin film sample was calculated using the following equation:

$$\sigma = \frac{1}{R_{ion}} \frac{1}{(N-1)l} \frac{d}{h} \tag{2}$$

Where $R_{ion}$ is the ionic resistance, $d = 8\,\mu m$ is the spacing between adjacent electrode teeth, $l = 500\,\mu m$ is the length of the electrode, $N = 80$ is the number of electrodes, $h$ is the thickness of the film.

## Thermalgravimetric Analyzer (TGA)

TGA measurements were conducted on a TA Instruments Discovery thermogravimetric analyzer at a scanning rate of 20 °C min⁻¹ within the temperature range from room temperature to 600 °C. The TGA measurements were performed under $N_2$ environment. The results are shown in Supplementary Figs. 18–19.

## Differential Scanning Calorimetry (DSC)

DSC measurements were conducted on a TA Instruments Discovery 2500 at a scanning rate of 10 °C min⁻¹. The sample was kept under a nitrogen environment during the experiment. The scan range for neutral PBBNB is from 0 °C to 300 °C. (see Supplementary Fig. 20a). For charged PBBNB⁺Br⁻, a needle was used to poke a small hole on an aluminum hermetic DSC pan and the sample was held at 180 °C to remove the absorbed water from ambient environment under nitrogen flow (see Supplementary Fig. 20b) and then, the sample was scanned from 0 °C to 250 °C. The same sample was then scanned from 0 °C to 260 °C to confirm the degradation peak at around 250 °C (see Supplementary Fig. 20c). Another charged PBBNB⁺Br⁻ was held in the hermetic DSC pan and was pretreated in a 95% RH chamber with hermetic lid open. The sample was equilibrated in the humidity chamber for two hours. Hermetic pan was then closed inside the humidity chamber to reserve absorbed water molecules by cationic functional group. DSC measurements were performed to the sample ranging from 0 °C to 95 °C (see Supplementary Fig. 20d).

## Small-Angle X-Ray Scattering (SAXS)

SAXS experiments were performed with SAXSLAB (XENOCS)'s GANESHA at University of Chicago. SAXS experiments were performed on polymer powder samples. The results are shown in Supplementary Fig. 24.

## In situ IR and 2D IR

In situ IR absorption and 2D IR measurements were performed on 50 μm films that were spincoated onto a $CaF_2$ window. The polymer sample was held in a demountable flow cell (Harrick) with a spacer between the polymer and the front window of the cell to allow for the flow of humidity-controlled air over the sample. The RH controlled air was generated with a humidity generator (RH95, Linkam) with the humidity sensor placed in line before the sample. For FTIR measurements the RH was stepped from 5% to 75% in 5% steps and allowed to equilibrate for 10 min before the measurement of the spectrum. Dry N2 was flowed over the sample to obtain the 0% RH spectrum, which was used as a reference spectrum to subtract the background from the polymer. IR absorption measurements were performed in an FTIR spectrometer (Bruker, Tensor 70).

Polarization dependent IR transient absorption (TA) and 2D IR measurements were performed on the same sample with the RH stepped from 25% to 75% in steps of 10%. Experiments were performed using a homebuilt spectrometer equipped with a 4-f pulse shaper with a Ge AOM (Phasetech QuickShape) and operating in the pump-probe geometry. We used a Ti:Sapphire regenerative amplifier (Spectra-Physics Solstice, 800 nm, 3 mJ per pulse energy, 1 kHz, 90 fs pulse duration) to pump an optical parametric amplifier equipped with difference frequency generation (Light Conversion TOPAS Prime). The resulting mid-IR pulse was centered at 2530 cm⁻¹ with 100 fs duration with 17 μJ per pulse. The probe beam was split from the pump by the front face reflection off an uncoated wedged $CaF_2$ window. The waiting time $t_2$ was controlled with a delay stage in the probe line (Aerotech).

The pump was passed through the pulse shaper, which was used for temporal compression and to generate a pair of pulses with controllable time delay $t_1$ (0 to 3 ps in 25 fs steps, rotating frame at 1500 cm⁻¹, 2 × 2 phase cycling) for 2D IR measurements. The pump was compressed by optimizing the second harmonic generation in AGS as a function of the 2nd, 3rd and 4th order dispersion applied by the pulse shaper and verified by measuring the interferometric autocorrelation.

The pump and probe pulses were focused into the sample with a 4 inch focal length 90⁰ off-axis parabolic mirror and the change in absorption of the probe induced by the pump was determined using a spectrograph (Horiba Triax 190) and detected with a 64 pixel HgCdTe photodiode array (Infrared Associates MCT-7-64, Infrared Systems Development IR-6416) to obtain the detection frequency axis $\omega_3$.

The polarization dependence of the TA and 2D IR measurements were determined by rotating the polarization of the pump to 45⁰ relative to the probe using a $\lambda/2$ waveplate and rotating the analyzer in front of the spectrometer between +45⁰ (parallel signal) and −45⁰ (perpendicular signal) on sequential scans. The isotropic signal was determined from the parallel and perpendicular signals as $S_{iso} = (S_{\parallel} + 2S_{\perp})/3$. The anisotropy was determined as $r = (S_{\parallel} - S_{\perp})/S_{iso}$. For 2D IR measurements we used the isotropic signal. The results are shown in Supplementary Figs. 33–35.

## Simulation methods

All molecular dynamics (MD) simulations use an all-atom force field, with parameters for bromide ions and quaternary ammonium groups from the Canongia Lopes & Padua (CL&P)[48] force field, and all other parameters from the All-Atom Optimized Potentials for Liquid Simulations (OPLS-AA)[49] force field. In addition, the SPC/E water model is used in hydrated systems. All MD simulations are performed by using the GROMACS[50] 2022.4 simulation package. The equations of motion are evolved by utilizing the leap-frog algorithm with a 1 fs timestep. A cutoff distance of 12 Å is used to calculate nonbonded interactions while smooth particle-mesh Ewald (SPME) method is used to compute longer-range electrostatics. The v-rescale thermostat and Berendsen barostat are used for initial equilibration in NPT simulations. The Nosé-Hoover thermostat and Parrinello-Rahman barostat are employed for further NPT simulations. Production simulations with NVT ensemble are performed with Nosé-Hoover thermostat. Additional details of the simulation protocols and parameters are provided in the

supplementary information (see Supplementary Fig. 25 and Supplementary Tables 2–6).

## Data availability

All data are available within the main text and the supplementary materials.

## Code availability

The MD simulations were conducted using the commercially available package Gromacs[50]. Details required to reproduce the computations and the code supporting the key findings of this study have been provided in the "Methods" section and Supplementary Information. Additional codes used in this study are available from the corresponding authors upon request.

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

## Acknowledgements

This work was supported by the Department of Energy, Office of Basic Energy Sciences, Division of Materials Science and Engineering. The authors gratefully thank Dr. Xiaoying Liu for her assistance in performing DVS measurements, and Dr. Philip Griffin for his diverse assistance throughout this study. This work made use of the Pritzker Nanofabrication Facility part of the Pritzker School of Molecular Engineering at the University of Chicago, which receives support from Soft and Hybrid Nanotechnology Experimental (SHyNE) Resource (NSF ECCS-2025633), a node of the National Science Foundation's National Nanotechnology Coordinated Infrastructure. Parts of this work were carried out at the Soft Matter Characterization Facility of the University of Chicago.

## Author contributions

S.N.P., P.F.N., and J.J.d.P. conceived the project. Z.W. completed sample preparations, performed, DSC, TGA, SEC, NMR, FTIR, EIS, analyzed and oversaw all experimental data. M.K. conducted part of DSC experiments and performed SAXS experiments. G.S. and J.J.d.P. designed the model and performed the simulations. A.S. and G.S. applied the graph theory. N.H.C.L. and A.T. performed in situ IR and 2D IR experiments. M.M. and P.A.K. synthesized PBBNB polymer. K.W. fabricated IDEs. A.T. and A.B.M. helped with in situ Ellipsometer. J.M.M.d.O. contributed to the calculation of the potential of mean force. All authors participated in manuscript preparation and editing.

## Competing interests

The authors declare no competing interests.
