## [Transparent Peer Review file · Nature Communications]

Water-Mediated Ion Transport in an Anion Exchange Membrane

Corresponding Author: Professor Juan de Pablo

Version 0:

Reviewer comments:

Reviewer #1

(Remarks to the Author)

The manuscript provides detailed analysis of water and ion transport in anion-exchange membrane using a set of experimental and atomistic simulation analysis. The manuscript focuses on Br⁻ anion exchange mechanism and correlates the ionic mobility with rotational and translational motion of water in ion coordination shell. Overall, the analysis and data presented for this specific system are very strong, the proposed mechanism is new and supported by the presented data. The manuscript is well-written and results are well illustrated by the data. But I have couple concerns:

1) Majority of AEMs that has been considered so far involve OH⁻ transport which in turn involves Grotthuss mechanism in addition to vehicular. There were a number of modeling and experimental papers showing that that Grotthuss mechanism is important in polymeric AEMs particularly at low hydration levels where there is a substantial degree of confinement of hydrated anions. Br⁻ conducting AEMs are less common and my understanding do not involve the Grotthuss mechanism for conductivity. Hence the presented analysis and mechanism might not be valid or relevant for all AEMs as the title and abstract imply. If this is a generic mechanisms it has to be demonstrated on hydroxide- conducting membranes. if that is not an option, it should be clearly stated that it is applicable for this type of AEMs

2) I am not sure why reference 6, which is primarily talks about proton conducting membranes and Nafion is used as representative reference on current knowledge for Anion-exchange membranes. There is plenty of literature including modeling on AEMs and mechanisms of transport which is not included.

In summary, the presented experimental and simulation data support the proposed mechanism and it is new and interesting mechanism. However, the reviewer is not sure how generic this mechanism for all AEMs particularly those that involve significant contributions from the Grotthuss mechanism, e.g. in OH⁻ conducting membranes.

Reviewer #2

(Remarks to the Author)

In the last couple of decades it was demonstrated that the overall ion conductivity of condensed phases is originated from the superimposition of the contributions originating from at least two ion conductivity pathways, associated respectively to: (i) the migration of ions between delocalization bodies (DBs); and (ii) the migration of ions along one or more interdomain pathways. For the sake of example, in hydrated ionomers there are one or more interdomain pathways along the interfaces between hydrophobic and hydrophilic domains (giving rise to σ_{IP} "interdomain" conductivity), together with a pathway associated with ion migration between DBs (giving rise to σ_{EP} "electrode polarization" conductivity). See for example: J. Am. Chem. Soc., 134, 19099-19107 (2012).

It is pointed out that this framework of interpretation is general, covering very diverse materials including anion-exchange ionomers (e.g., J. Am. Chem. Soc. 140, 1372-1384 (2018)), cation-exchange ionomers (e.g., J. Am. Chem. Soc. 142, 801-814 (2020)), and also other ion-conducting media such as hybrid inorganic-organic polymer electrolytes for application in lithium-ion batteries (e.g., Int. J. Hydrogen Energy, 39, 2884-2895 (2014)). This framework of interpretation has been widely discussed in the literature in the last couple of decades.

In the present paper, the authors simply confirm previous findings concerning the mechanism of ion conduction. On this basis, the present paper only offers an incremental progress and does not exhibit the breakthrough character that is required to warrant publication in this Journal. Hence, the present manuscript must be rejected.

Specific points

- The authors should not insist on anachronistic and inconsistent “Grotthus” and “vehicular” long-range conductivity pathways for ionomeric materials.
- Page 3, Line 53. The authors must clarify the origin of the statement “Ion transport and water structure are highly coupled in an AEM”.
- Page 4, Lines 70-74. There is no comparison with other vibrational spectroscopy studies (see for instance: J. Phys. Chem. B, 113, 632-639 (2009); Solid State Ionics, 252, 84-92 (2013); Phys. Chem. Chem. Phys. 17, 4367-4378 (2015); and Phys. Chem. Chem. Phys. 21, 10357-10369 (2019)).
- Page 5, Lines 80-82. The phenomena described by the authors actually aren't associated to the “vehicular” transport mechanism, but to the σ_{EP} conductivity.
- Page 5, Lines 84-86. The temperature range explored by the authors is too narrow to support the conclusion that the $1000/T$ plot actually follows an Arrhenius behavior.
- Page 5, Lines 91-93. This statement is not correct. As reported elsewhere (Chem. Mater. 32, 38-59 (2020)), the overall conductivity is originated from the superimposition of the contributions arising from at least two ion conductivity pathways, i.e., σ_{IP} and σ_{EP} . At a low relative humidity, in ionomers the dominant conductivity pathway is σ_{IP} . At a high relative humidity, the dominant conductivity pathway is σ_{EP} .
- Page 5, Lines 93-96. It is difficult to admit that at $\lambda = 14$ the water is able to flow through the ionomer.
- Page 7, Lines 121-123. This weak dissociation is actually responsible for another, different conductivity mechanism (i.e., likely a contribution to σ_{IP}).
- Page 9, Lines 153-155. The authors must reconsider their interpretation also taking into account previous work discussing the vibrational spectra of hydrated ionomers, including: (see for instance: J. Phys. Chem. B, 113, 632-639 (2009); Solid State Ionics, 252, 84-92 (2013); Phys. Chem. Chem. Phys. 17, 4367-4378 (2015); and Phys. Chem. Chem. Phys. 21, 10357-10369 (2019)).
- Page 10, lines 173-177. The phenomena reported here are actually described elsewhere as “delocalization bodies” J. Am. Chem. Soc., 134, 19099-19107 (2012); and J. Am. Chem. Soc. 140, 1372-1384 (2018)).
- Page 11, lines 195 and following. The authors must describe how the results of their simulation can be interpreted on the basis of the conductivity pathways σ_{IP} and σ_{EP} .
- Page 12, lines 208-208. The authors must describe how this statement fits with the two conductivity pathways σ_{IP} and σ_{EP} described elsewhere.
- Page 15, lines 261-271. This is simple speculation, without any experimental evidence. It is impossible that these processes, which occur at the 100-400 ps timescale, are the rate-determining steps of the conductivity which, as stated in Fig. 1, takes place with relaxation times longer than 1 microsecond.
- Page 15, lines 277 and following. This statement is not clear.
- Page 16, lines 279-281. This is really confusing. The authors are comparing the activation energy E_a determined on the long-range migration processes with relaxation times longer than 1 microsecond with local motions with relaxation times on the order of picoseconds.
- Page 16, lines 291 and following. These two interpretations are not new; they have already been extensively described in the literature also including results obtained by means of broadband electrical spectroscopy (BES). The results reported in this work do not detect any new event, but simply confirm findings widely accepted in the literature and interpreted on the basis of two (or more) conductivity pathways contributing to the overall conductivity of ionomers ($\sigma = \sigma_{IP} + \sigma_{EP}$)

Reviewer #3

(Remarks to the Author)

Overall Evaluation: The manuscript addresses a significant and timely issue in the field of ion exchange membranes (AEMs) with respect to water clustering and dynamics. The use of advanced two-dimensional infrared spectroscopy (2DIR) and semiclassical simulations offers a comprehensive view of the water structure and its impact on ionic transport. The findings

are novel, presenting a dual-regime mechanism of ion transport influenced by the reorientation dynamics of water molecules in solvation shells. This study bridges a crucial gap in understanding the molecular basis of ion transport in AEMs, which has implications for various electrochemical applications. The work is methodologically sound, and the results are promising for the future design of next-generation AEMs.

Key Strengths:

1. Innovative Approach: The application of 2DIR spectroscopy combined with semiclassical simulations to study water and ion dynamics in AEMs is groundbreaking. The theoretical methodologies, particularly the water clustering analysis, provide an unprecedented view of the molecular mechanisms governing ionic transport.
2. Comprehensive Analysis: The study provides a thorough analysis of water and ion dynamics across different humidity regimes, contributing to a deeper understanding of the relationship between solvation structure and ion conductivity.
3. Detailed Characterization: The characterization of the water network and its influence on bromide ion transport is detailed and well-supported by experimental data.

Key Issue to be Addressed:

The manuscript presents an excellent study; however, there is a critical point that needs to be addressed to enhance the clarity and robustness of the findings:

Justification of Conductivity Predictions:

The MD simulation study accounts for vehicular ion diffusion but does not consider the Grotthuss hopping mechanism, which is commonly known to play a significant role in ion conductivity. Despite this, the simulated conductivity matches the experimental values very closely, as provided in Figure 26 in the SI. The authors need to provide a robust justification for how their classical molecular dynamics (MD) calculations can accurately predict ion conductivity without incorporating the Grotthuss hopping mechanism. This is particularly important given the significant role that Grotthuss hopping is known to play in ionic conductivity (e.g., <https://doi.org/10.1021/acs.jpcc.8b02318>). In fact, I would like to know how do the authors reconcile the exclusion of Grotthuss hopping with the close quantitative match between the calculated and experimental conductivity values? Could there be a complementary mechanism or an overlooked aspect in the current simulation that compensates for the absence of Grotthuss hopping?

Also, I believe the authors should highlight the potential limitations of the current simulation approach, and how might they affect the interpretation of the results?

One very minor point is that in Figure 3h in the manuscript, if the y-axis shows the probability density, the unit should be ns^{-1} .

Final Recommendation: The manuscript presents a significant contribution to the field and is worthy of publication in Nature Communications, provided that the authors address the key issue regarding the justification for their conductivity predictions and highlight the limitation of the current methodology in providing the full picture for ion transport in AEMs. With this clarification, the paper will provide a robust and comprehensive understanding of water and ion dynamics, paving the way for future advancements in membrane technologies.

Reviewer #4

(Remarks to the Author)

In this paper, information about the state and dynamics of water is derived from the results of MD. In AEM, the polarized parts of the polymer form a first hydrated structure, and as the water content increases, a second hydration is formed that connects to the water network in the polymer, increasing the ionic conductivity. This general conjecture about the ionic conductivity of AEM is described in S-Figure 1 of this paper. This paper carefully verifies and supports this concept using 2D-IR and MD. Although the results obtained are rather within the range of expectation, it is a very interesting paper in that it is clarified by careful MD simulations and experiments. On the other hand, although a comment is made in the conclusion about the ideal molecular structure of AEM that realizes fast ionic conductivity, it cannot be said that the results are particularly new. In this paper, water is studied and there is no discussion of the influence of the polymer or how the type of ion or the strength of the interaction affects the mechanism. Although the polymer certainly moves slowly and can be omitted from the discussion, it affects the dispersion of the water molecules and must be considered in the molecular design of the AEM. In conclusion, however, the paper is based on thorough experiments and in-depth considerations based on comparisons with MD, and may be considered for publication in Nature Communications.

The following are comments should be addressed.

1. MD results depend on the initial configuration. Especially in the low RH region, the dispersion of water molecules may not be reproducible. Since it seems that only one model is considered in this paper, it is desirable to study and compare results for different initial configuration to strengthen the validity in the low RH region.
2. Information on the number of water molecules in the model is needed. Also, information on density or cell length would be useful to researchers performing similar calculations.
3. Fig.2. c: It is necessary to mention that the RH dependence of CN is different from a and b.
4. Fig.3 h: Although the distribution of time to cross a distance of 1 nm is shown, there are likely many molecules that

frequently enter and exit the boundary. The effect of these may be large and change depending on the sampling interval.

5. Fig.3: Is the k related to the hydration number? If so, please discuss it together with CN in Fig.2.

Version 1:

Reviewer comments:

Reviewer #1

(Remarks to the Author)

I believe the revised version of the manuscript addressed all the comments and can be published in current form.

(Remarks on code availability)

Reviewer #2

(Remarks to the Author)

Though the authors have addressed several of the concerns raised by the reviewer, there are still several fundamental points that were not considered properly, pointing to a severe misunderstanding of the phenomena associated to the ion transport phenomena. For these reasons, to prevent the spread of misleading information in the literature, the manuscript cannot be accepted for publication and must be rejected.

The points not addressed properly by the authors of the manuscript are the following:

- The ion-transport mechanism involving different conductivity pathways can also be applied to homogeneous systems that do not exhibit the clear phase separation of hydrated perfluorinated ionomers such as hydrated Nafion. The ion-transport mechanism involving different conductivity pathways has also been applied to homogeneous systems such as ionic liquids (see for instance: J. Power Sources 565, 232910 (2023)). The different domains along whose boundaries the charge can accumulate need not correspond to different hydrophilic/hydrophobic domains. They may simply correspond to different domains (e.g., areas where the packing of the macromolecules is only slightly different) within an otherwise homogeneous system (e.g., the polynorbornene homopolymer considered by the authors). This point was not addressed at all in the revised version of the manuscript, and incorrectly leads the authors to not apply the well-established s(EP+IP) model to their system.
- If the temperature range is small enough, the ion conductivity any system can be approximated with an Arrhenius trend. At the very least this point should have been mentioned clearly in the revised text, including the corresponding justification for such a gross approximation. In addition, the absence of a clear T_g at $T > 0^\circ\text{C}$ (the range measured by the authors) may simply indicate that the T_g actually occurred below 0°C . Thus, there is no proof that prevents the proposed materials to follow a VTF conductivity trend, leading to incorrect analysis and conclusions.
- Finally, the authors have not considered appropriately the issue of different timescales (water vs. the entire system) owing to their reluctance to consider appropriately the issue of multiple conductivity pathways in an apparently homogeneous system.

(Remarks on code availability)

N/A

Reviewer #3

(Remarks to the Author)

After reviewing the revisions and your clarifications, I am fully convinced that your paper meets the necessary standards for publication in Nature Communications and I am pleased to recommend it for publication in its current form.

(Remarks on code availability)

Reviewer #4

(Remarks to the Author)

In response to my comments, this answer is sufficient. I recommend that the current state of the solution is worthy of publication.

(Remarks on code availability)

Version 2:

Reviewer comments:

Reviewer #3

(Remarks to the Author)

Regarding the responses the authors provided to the question of reviewer #2, particularly about the assumption of a “homogeneous” system in which the dynamics of the water and ions are markedly faster than the dynamics of the polymer, I believe the answers provided are largely valid.

The authors correctly noted that the T_g of this polymer is above 300 °C (based on the references provided from the literature); however, their reference to the extremely slow dynamics of the polymer and side chain in simulations is not entirely accurate. Although such slow dynamics compared to those of the water and ions support their argument—and I believe this should be noted in the paper—it does not prove that the actual T_g of the polymer is above the simulation temperature. In fact, due to the extremely smaller time frame of simulations compared to experimental tests (such as DSC), even soft polymers can exhibit very limited segmental motions in atomistic simulations.

To reconcile the true concern of the reviewer with the response of the authors, I think the authors could soften the tone about the homogeneity of the polymer while arguing that although it is an assumption, it is a valid and robust one, as there is no evidence of crystallinity, phase separation, or ordered/disordered domains reported or found for this membrane. Additionally, the negligible dynamics of polymer compared to water and ion, as evidenced from both DSC measurements and MD simulations (I would suggest the inclusion of mean squared displacement curves for the atoms of the polymer backbone and sidechain and water molecule in one graph to be comparable), provides extra support for the justification of the conclusions of the paper.

(Remarks on code availability)

Response to Reviewers' Comments

Manuscript ID: NCOMMS-24-35031-T

Manuscript Title: Water Dynamics, Water Structure, and Ion Transport in an Anion Exchange Membrane

We would like to thank the reviewers for taking the time to provide their thorough and insightful comments. We have carefully considered each comment and have revised the manuscript accordingly. Our point-by-point response to specific comments and the changes made are detailed below. Page and paragraphs (**para.**) numbers quoted below refer to those in the marked manuscript file (with yellow highlighter).

Reviewer #1

The manuscript provides detailed analysis of water and ion transport in anion-exchange membrane using a set of experimental and atomistic simulation analysis. The manuscript focuses on Br- anion exchange mechanism and correlates the ionic mobility with rotational and translational motion of water in ion coordination shell. Overall, the analysis and data presented for this specific system are very strong, the proposed mechanism is new and supported by the presented data. The manuscript is well-written and results are well illustrated by the data. But I have couple concerns:

We thank the reviewer for the thorough and insightful comments.

1. Majority of AEMs that has been considered so far involve OH- transport which in turn involves Grotthuss mechanism in addition to vehicular. There were a number of modeling and experimental papers showing that that Grotthuss mechanism is important in polymeric AEMs particularly at low hydration levels where there is a substantial degree of confinement of hydrated anions. Br-conducting AEMs are less common and my understanding do not involve the Grotthuss mechanism for conductivity. Hence the presented analysis and mechanism might not be valid or relevant for all AEMs as the title and abstract imply. If this is a generic mechanisms it has to be demonstrated on hydroxide- conducting membranes. if that is not an option, it should be clearly stated that it is applicable for this type of AEMs

Response: We appreciate the reviewer's comments and agree with their assessment. Our study indeed focuses on Br- conducting AEMs which do not involve the Grotthuss mechanism for conductivity. We have chosen to study these AEMs for several reasons:

(1) Applications of AEMs Conducting Halide Ions: AEMs that conduct halide ions have various applications, such as electrodialysis for desalination and brine treatment, chlor-alkali processes in an electrolyzers, water treatment and purification, and bromine-based flow batteries. Understanding the fundamental mechanisms of Br- conducting AEMs can provide valuable insights into the molecular design of these membranes, with significant implications for these electrochemical applications.

(2) Insights into OH⁻ Conducting AEMs: Investigating Br- conducting AEMs allows us to exclude the Grotthuss mechanism and focus solely on other potential mechanisms. This is particularly

useful given the ongoing debate in the literature regarding the relative contributions of Grotthuss and vehicular mechanisms in OH⁻ conducting AEMs ((J. Phys. Chem. Lett. 2018, 9, 825– 829, DOI: 10.1021/acs.jpcllett.8b00004, J. Am. Chem. Soc. 2016, 138, 991– 1000 DOI: 10.1021/jacs.5b11951). By focusing on bromide ion transport, we intentionally excluded the Grotthuss mechanism to thoroughly investigate the effects of water dynamics, water structure, and ion solvation in ion transport without the influence of proton hopping. We believe that a clear understanding of these effects will provide insights and inform our future computational studies on OH⁻ conducting AEMs.

To study OH⁻ transport, advanced quantum mechanical methods, such as *ab initio* MD and QM/MM, and Machine-Learning Potentials can be employed. Machine-Learning Potentials, in particular, show significant promise due to their speed and quantum-level accuracy in systems involving bond breaking and forming, enabling the capture of the Grotthuss mechanism. Our group has recently developed effective Machine-Learning Potentials, which will be employed in a future paper on OH⁻ transport.

By combining these advanced methods with classical MD simulations, we will be able to achieve a comprehensive understanding of OH⁻ conducting AEMs. To further address the referee's concerns, we have revised the abstract to clarify that our findings are relevant specifically to Br⁻ conducting AEMs.

Changes to the manuscript: We have revised the abstract for better clarification.

Our findings provide molecular-level insights into AEMs with inherent transport of halide ions, and help pave the way towards a comprehensive understanding of hydroxide ion transport in AEMs.

We have also added discussions in the introduction for better clarification.

2. *I am not sure why reference 6, which is primarily talks about proton conducting membranes and Nafion is used as representative reference on current knowledge for Anion-exchange membranes. There is plenty of literature including modeling on AEMs and mechanisms of transport which is not included.*

Response: We agree with the referee, and we have updated our references to include more relevant references specifically focusing on AEMs. We have cited key references that represent the current knowledge and modeling of AEMs.

Changes to the manuscript: We added the following references to the manuscript:

Recent studies on anion-exchange membranes (AEMs), including both experimental and modeling approaches, have significantly advanced our understanding of water absorption and ion transport mechanisms⁶⁻⁸.

6. R. Varcoe, J. *et al.* Anion-exchange membranes in electrochemical energy systems. *Energy Environ. Sci.* **7**, 3135–3191 (2014).

7. Arges, C. G. & Zhang, L. Anion Exchange Membranes' Evolution toward High Hydroxide Ion Conductivity and Alkaline Resiliency. *ACS Appl. Energy Mater.* **1**, 2991–3012 (2018).

8. Yang, Y. *et al.* Anion-exchange membrane water electrolyzers and fuel cells. *Chem. Soc. Rev.* **51**, 9620–9693 (2022).

In summary, the presented experimental and simulation data support the proposed mechanism and it is new and interesting mechanism. However, the reviewer is not sure how generic this mechanism for all AEMs particularly those that involve significant contributions from the Grotthuss mechanism, e.g. in OH⁻ conducting membranes.

Response: We appreciate the reviewer's feedback. Disentangling the contributions of Grotthuss and vehicular mechanisms has been a long-lasting challenge in the study of OH⁻ transport. A shared aspect of both Grotthuss and vehicular mechanisms is that water dynamics, water structure, and ion solvation all contribute to ion transport. We intentionally exclude the Grotthuss mechanism to thoroughly investigate water dynamics, water structure, and ion solvation without the influence of proton hopping. We believe that without understanding these effects, it will be challenging to probe OH⁻ transport directly. We believe our methodology (EIS, MD, and ultra-fast 2D IR) offers an effective way to understand OH⁻ transport in future work. For studying OH⁻ transport, other computational methods, particularly quantum mechanical approaches such as *ab initio* MD and quantum mechanics/molecular mechanics (QM/MM), as well as newer approaches based on Machine Learned (ML) potentials, are necessary to capture proton hopping mechanisms. By combining these quantum mechanical/ML methods with classical MD simulations, we will be able to achieve in future work a comprehensive understanding of OH⁻ conducting AEMs.

Changes to the manuscript: We added the following discussion to the introduction on page 4.

As explained above, a key aspect shared by both the Grotthuss and vehicular mechanisms is the integral role of water dynamics, water structure, and ion solvation in ion transport. By focusing on bromide ion transport, we intentionally excluded the Grotthuss mechanism to thoroughly investigate these factors without the influence of proton hopping. We believe that a clear understanding of these effects is important before directly probing OH⁻ transport.

We added a discussion on the future directions for our simulation approach on page 17.

For study of OH⁻ transport, other computational methods, particularly quantum mechanical approaches such as *ab initio* MD and quantum mechanics/molecular mechanics (QM/MM), or alternative approaches based on machine learning (ML) potentials, are necessary to capture proton hopping mechanisms. By combining these quantum mechanical and ML methods with classical MD simulations, we hope to achieve in future work a comprehensive understanding of OH⁻ conducting AEMs.

Reviewer #2

*In the last couple of decades it was demonstrated that the overall ion conductivity of condensed phases is originated from the superimposition of the contributions originating from at least two ion conductivity pathways, associated respectively to: (i) the migration of ions between delocalization bodies (DBs); and (ii) the migration of ions along one or more interdomain pathways. For the sake of example, in hydrated ionomers there are one or more interdomain pathways along the interfaces between hydrophobic and hydrophilic domains (giving rise to σ_{IP} “interdomain” conductivity), together with a pathway associated with ion migration between DBs (giving rise to σ_{EP} “electrode polarization” conductivity). See for example: *J. Am. Chem. Soc.*, 134, 19099-19107 (2012).*

*It is pointed out that this framework of interpretation is general, covering very diverse materials including anion-exchange ionomers (e.g., *J. Am. Chem. Soc.* 140, 1372-1384 (2018)), cation-exchange ionomers (e.g., *J. Am. Chem. Soc.* 142, 801-814 (2020)), and also other ion-conducting media such as hybrid inorganic-organic polymer electrolytes for application in lithium-ion batteries (e.g., *Int. J. Hydrogen Energy*, 39, 2884-2895 (2014)). This framework of interpretation has been widely discussed in the literature in the last couple of decades.*

In the present paper, the authors simply confirm previous findings concerning the mechanism of ion conduction. On this basis, the present paper only offers an incremental progress and does not exhibit the breakthrough character that is required to warrant publication in this Journal. Hence, the present manuscript must be rejected.

Response: We thank the referee for pointing out these important papers on the thermophysical and ion transport properties of polyelectrolytes. The papers mentioned by the reviewer were conducted by Dr. Vito Di Noto and his group. Their studies, which combined high-resolution thermogravimetry (HR-TG), modulated differential scanning calorimetry (MDSC), dynamic mechanical analysis (DMA) and broadband electrical spectroscopy (BES), have contributed to establishing fundamental relationships between thermophysical properties and ion transport in polyelectrolytes.

Note, however, that our work is different. It addresses fundamentally different aspects of ion transport. Specifically, we study water structure and water dynamics simultaneously, a feature that has not been addressed in the studies mentioned above. Additionally, we examine ion transport at various scales, and its cooperativity with water, thereby providing a more detailed and nuanced understanding. These new insights go well beyond the existing findings, and they advance the understanding of ion transport in polyelectrolytes. We believe our approach has contributed much needed complementary, rather than overlapping information, vis-a-vis the work presented in the referenced papers.

The reviewer states that ion transport in condensed phases can be represented by at least two ion transport pathways: (i) the migration of ions between delocalization bodies (DBs); and (ii) the migration of ions along one or more interdomain pathways. This theory is applicable to Nafion[®] ionomer and was described in *J. Am. Chem. Soc.*, 134, 19099-19107 (2012). Nafion[®] ionomer is well known to be a semi-crystalline material, and it shows a phase-separated morphology, as evidenced by small angle X-ray scattering (SAXS) and wide-angle X-ray scattering (WAXS). The

PTFE backbone of Nafion[®] tends to form a crystalline structure, which subsequently affects the molecular packing of the sulfonic acid groups in the side chains. Ion transport in Nafion[®] is affected by the ratio of the crystalline/amorphous domains. (see *Langmuir* 2020, 36, 3871–3878) If we regard a crystalline domain as a grain, then ion transport within a grain and between each grain is different. The long-range ion transport will be limited by grain boundaries. The effects of grain boundaries and grain size on ion transport are well understood in the solid-state polymer electrolyte community. (see *Macromolecules* 2014, 47, 5424–5431; *Energy Sci Eng.* 2022;10:1643–1671). The reviewer discusses another example in *J. Am. Chem. Soc.* 140, 1372–1384 (2018) to show that the two-ion transport pathway theory can be applied to anion exchange membrane (AEM) materials. The AEM material system is lamellar-forming poly(vinyl benzyltrimethylammonium)-b-poly(methylbutylene) with different ion exchange capacities (IECs). This is a block copolymer material with long-range periodic order. Ion transport happens within the ion conducting layer and across different ion conducting layers because the lamellar-forming materials exhibit inherent defects between each layer. (see *Mol. Syst. Des. Eng.*, 2019, 4, 519; *Nano Lett.* 2019, 19, 4684–4691) The ion transport across the ion conducting layer and within ion conducting layer will be different and can be observed by the BES technique. Using BES to probe these phenomena is a convenient approach to quantify ion transport behaviors between different DBs.

One commonality between the Nafion[®] study and the above AEM study is that both material systems exhibit a nano- or micro-phase-separated morphology, and ion transport is different within the ion conducting domain (interdomain) and across the ion conducting domain (delocalized body, DB). Thus, ion transport can be nicely captured by the theory proposed by the author. However, in our study, we consider a totally different material system. The polynorbornene homopolymer has randomly distributed hydrophilic side chains and a hydrophobic polymer backbone. The material is homogeneous. There is no phase separation, no semi-crystalline structure, or ion clusters in the polymer, as evidenced by SAXS data in the supplementary Fig. S24. Thus, we do not believe the two ion transport pathway theory can be applied to our system.

Importantly, our combined 2D IR experimental and simulation molecular-level approach is entirely different from that presented in past studies and has never been explored as a means to study ion transport in hydrated polyelectrolytes as a function of relative humidity. We want to highlight the following **innovations** of our work:

- i. We are able to probe water dynamics, H-bonding dynamics, and ion transport in a state-of-the-art AEM chemistry, specifically polynorbornene, from femtoseconds (fs) to milliseconds (ms) using ultra-fast 2D IR, atomistic molecular dynamics (MD), and electrochemical impedance spectroscopy (EIS). We validated the accuracy of our MD models by comparing experimental ionic conductivity and water volume fraction data as a function of relative humidity with our MD results. We achieve excellent agreement between both the EIS data and MD data, and the ultra-fast 2D IR data and MD data. The methodology that precisely captures water dynamics, H-bonding dynamics and ion transport has never been done before.
- ii. In our study, we aim to investigate ion transport, with an emphasis on the fast mechanism in an AEM. We are able to observe the transition between the slow transport mechanism and fast transport mechanism from EIS, MD, and ultra-fast 2D IR, which has not been done before. From the description of vehicular mechanism, it requires free, also known as unbound, water within the membrane (*Nat Energy* 6, 339–348 (2021)). However, water molecules are never able to fully

dissociate Br⁻ from the quaternary ammonium, and there is no free water in our materials. This new understanding has not been explored by the scientific community before. Moreover, by investigating the structure of water in the system, we provide a quantitative description of the required water percolation network at a molecular level that enables faster ion transport events. We find that the faster ion transport mechanism is enabled by the formation of a robust water network with at least 3 edges ($k = 3$) (please see the full definition of the “edge” described by graph theory in our manuscript), and a large population of water molecules in the 2nd solvation shell that reduce the bridging effect and the constraints on the ions. The new ion transport mechanism has not been described in the papers mentioned by the referee, and the molecular level understanding presented in our study has never been introduced.

iii. We deployed ultra-fast 2D IR technology to probe water dynamics in the polynorbornene AEM system. We reach three conclusions that have not been clarified by the polyelectrolyte community. First, it is commonly believed that there exists bulk water, weakly polarized water, and strongly polarized water in an AEM. Based on our ultra-fast 2D IR results, we have demonstrated that the Br⁻ transport takes place in an environment that is highly confined by the polymer and does not resemble bulk-water. This is a key finding that helps us understand whether vehicular transport occurs in our system. Second, we have used ultra-fast 2D IR spectroscopy to investigate how the local solvation environment alters the electrostatic field, and thus affects water motions. For example, we have investigated and quantified how the water orientational dynamics affects the bottleneck of ion transport. Third, we have quantified and categorized water reorientation dynamics. We find that in low RH environments, the scarce water found in the Br⁻ second shell limits the mobility of the associated ions. At high RH environments, the water population in the second shell increases, diminishing the bridging effect and liberating ions from their previous constraints, leading to faster ion transport mechanism. All of these findings from ultra-fast 2D IR and the correlation between them and ion transport have not been investigated in polyelectrolyte community.

In summary, we have provided new and important molecular-level insights about ion transport in a state-of-the-art AEM material system. Not only are our findings original, but our 2DIR approach is novel and powerful, and will be pursued by others for the development of electrochemical systems using AEMs as separators.

Changes to the manuscript: We acknowledge these prior research efforts in the manuscript by adding the following discussion in **page 4-5**. We have now cited these papers in the main text.

Here we note that quasi-elastic neutron scattering (QENS)²³ has previously been used to examine water dynamics at picosecond timescales, and broadband electrical spectroscopy (BES) has been employed to investigate ion conducting pathways in polyelectrolytes²⁴⁻²⁶.

Specific points

- *The authors should not insist on anachronistic and inconsistent “Grotthuss” and “vehicular” long-range conductivity pathways for ionomeric materials.*

Response: We introduced the conventional concepts of Grotthuss and vehicular transport mechanisms in the introduction to highlight the importance of ion solvation, water dynamics, and water structure in understanding ion transport mechanisms. These mechanisms are still widely

discussed in recent literature (*Nat. Energy* **6**, 339–348 (2021), *Chem. Soc. Rev.* **51**, 9620–9693 (2022), *J. Phys. Chem. B* **126**, 2430–2440 (2022)) and are relevant to the current discourse in the field. Our research findings indicate that the vehicular transport mechanism alone cannot fully explain Br⁻ transport in the AEM. By presenting these traditional concepts, we were able to critically evaluate them and ultimately propose a new ion transport mechanism. Without this context, the development of our new insights would not have been possible.

Changes to the manuscript: We have provided additional explanation and introduction for these mechanisms on **pages 3-4**, in the same paragraph highlighted in the point below.

• *Page 3, Line 53. The authors must clarify the origin of the statement “Ion transport and water structure are highly coupled in an AEM”.*

Response: We agree with reviewer’s comment, and we have provided more explanation on this statement. This change is highlighted in **page 3-4**.

Changes to the manuscript:

Depending on the water absorption level, anion transport in an AEM is governed by three transport mechanisms: surface site hopping, vehicular transport, and Grotthuss hopping (OH-transport)^{3,8,11,12}. At low hydration levels, the hopping of anions between solvation sites that consist of cationic groups (e.g., N+, P+, and S+) is dominated by the surface site hopping mechanism¹³. The surface site hopping mechanism is correlated with the segmental mobility and solvation environment of the polymer chain^{14–16}. When the polymers are further exposed to humid environments, the vehicular and Grotthuss mechanisms begin to govern the overall anion transport. The vehicular mechanism involves concentration gradient-driven diffusion and electromigration, both dependent on the diffusion coefficient of ions moving through the membrane. In addition to vehicular transport, OH- transport is facilitated by the Grotthuss (proton hopping) mechanism^{11,17}. Water plays a crucial and complex role in these processes, including solvating anions, plasticizing polymers, forming hydrogen bonds (H-bonds) with anions, and clustering with other water molecules to form bulk-like configurations that enable vehicular diffusion and Grotthuss hopping (for OH- transport only)³. Overall, the discussion above serves to underscore that ion transport and water structure are tightly coupled in AEMs, and it is therefore critical to develop a detailed understanding of the molecular-level processes at play.

• *Page 4, Lines 70-74. There is no comparison with other vibrational spectroscopy studies (see for instance: *J. Phys. Chem. B*, 113, 632-639 (2009); *Solid State Ionics*, 252, 84-92 (2013); *Phys. Chem. Chem. Phys.* 17, 4367-4378 (2015); and *Phys. Chem. Chem. Phys.* 21, 10357-10369 (2019)).*

Response: We carefully reviewed the studies mentioned by the reviewer. All of these studies utilized Fourier Transform Infrared (FT-IR) spectroscopy, which is a one-dimensional linear spectroscopy technique (*Chem. Rev.* 2016, 116, 13, 7590–7607), to study the structures of various species. However, on Page 4, Lines 70-74, as we noted, “Ultra-fast two-dimensional infrared (2D IR) spectroscopy is used to measure the reorientation and fluctuations of water molecules,” which differs from FT-IR in several significant ways. Therefore, directly comparing our study with these

papers may not be entirely relevant. Nevertheless, we have reviewed and cited the relevant 2D IR studies, and to the best of our knowledge, this is the first use of 2D IR in the context of ion transport in hydrated polymers.

Changes to the manuscript: We added the following references in the manuscript.

19. Hamm, P. & Zanni, M. *Concepts and Methods of 2D Infrared Spectroscopy*. (Cambridge University Press, Cambridge, 2011). doi:10.1017/CBO9780511675935.

20. Reppert, M. & Tokmakoff, A. Computational Amide I 2D IR Spectroscopy as a Probe of Protein Structure and Dynamics. *Annu. Rev. Phys. Chem.* **67**, 359–386 (2016).

21. Khalil, M., Demirdöven, N. & Tokmakoff, A. Coherent 2D IR Spectroscopy: Molecular Structure and Dynamics in Solution. *J. Phys. Chem. A* **107**, 5258–5279 (2003).

22. Petti, M. K., Lomont, J. P., Maj, M. & Zanni, M. T. Two-Dimensional Spectroscopy Is Being Used to Address Core Scientific Questions in Biology and Materials Science. *J. Phys. Chem. B* **122**, 1771–1780 (2018).

• *Page 5, Lines 80-82. The phenomena described by the authors actually aren't associated to the "vehicular" transport mechanism, but to the σ_{EP} conductivity.*

Response: As summarized in Conclusions section, the faster ion transport mechanism in our system does not align with the vehicular mechanism. We introduced the conventional concepts of water clustering and vehicular transport on page 5 to provide context, but we later demonstrate that these do not apply to our system. Based on our MD and 2D IR analyses, there is no free water in our material, meaning that the vehicular mechanism cannot be used to explain the faster ion transport observed.

Regarding σ_{EP} conductivity, we believe it is unclear whether this concept can explain the faster ion transport in our system, given that our material is randomly distributed. Our material is not semi-crystalline and does not have nanostructures. Besides, our focus is on probing ion solvation, water dynamics, and water structures to understand ion transport mechanisms at the molecular level. We do not believe that the ($\sigma_{EP} + \sigma_{IP}$) theory will contribute to achieving our research objectives.

Changes to the manuscript: We have revised the discussions on **page 5** to eliminate any potential confusion.

The exponential increase in water uptake after 65% RH is generally attributed to water clustering. It is commonly believed that the formation of free water, also referred to as unbound water, is a prerequisite for the vehicular transport mechanism³. However, it remains unclear whether the formation of water clusters directly indicates the presence of free water. In the following sections, we examine in more detail the water structure and examine the underlying transport mechanisms in the context of our MD and 2D IR results.

• *Page 5, Lines 84-86. The temperature range explored by the authors is too narrow to support the conclusion that the $1000/T$ plot actually follows an Arrhenius behavior.*

Response: We believe our polymer exhibits Arrhenius behavior during the conductivity measurements. First, we measured the T_g of the dry polymer with DSC. The dry polymer does not show a T_g even up to 250 °C. We measured DSC for our polymer at 95% RH and still, there was no T_g detected during the measurement, indicating that our polymer remains in its glassy state during the whole conductivity measurements. The DSC results are shown in Supplementary Fig. 20. Second, our bond vector autocorrelation function (BVAf) results show that the reorientation dynamics of the polymer backbone and the pendant chain are very slow, confirming that the polymer is in glassy state. Therefore, the Arrhenius fit is the appropriate fitting model.

Regarding the temperature range, we conducted measurements from room temperature up to near 80 °C, which aligns with the normal operating conditions for fuel cells. Beyond this range, a change in state of water would occur, which are beyond the scope of interest in the field. Given these considerations, we believe the chosen temperature range is both appropriate and sufficiently broad, and the Arrhenius behavior is consistent with theoretical expectations.

• *Page 5, Lines 91-93. This statement is not correct. As reported elsewhere (Chem. Mater. 32, 38-59 (2020)), the overall conductivity is originated from the superimposition of the contributions arising from at least two ion conductivity pathways, i.e., σ_{IP} and σ_{EP} . At a low relative humidity, in ionomers the dominant conductivity pathway is σ_{IP} . At a high relative humidity, the dominant conductivity pathway is σ_{EP} .*

Response: On page 5 lines 91-93, we basically described the conductivity measurements for our materials, and we observed that the change of activation energy (E_a) falls into two regimes. We have not made any conclusion about conductivity pathways. Besides, we are interested in probing ion solvation, water dynamics, and water structures and understanding the ion transport mechanisms at a molecular level. We don't think the ($\sigma_{EP} + \sigma_{IP}$) theory will help us meet our research objectives. The polynorbornene homopolymer has randomly distributed hydrophilic side chain and hydrophobic polymer backbone. There are no phase separation, semi-crystalline structure, or ion clusters in the polymer. We are not sure if the theory is applicable to our material system.

• *Page 5, Lines 93-96. It is difficult to admit that at $\lambda = 14$ the water is able to flow through the ionomer.*

Response: We don't fully understand the reviewer's concern. A λ value of 14 is very common at high RH levels, such as 95%, in ionomers, as reported in many materials within this review paper (Chem. Rev. 2017, 117, 987–1104). At $\lambda = 14$, each quaternary ammonium is surrounded by 14 water molecules on average. On page 5, lines 93-96, we applied percolation theory to fit our conductivity data. Percolation theory is effective in explaining the transition from an ion insulator to an ion conductor, typically occurring at low RH levels for AEMs. However, it does not provide information at high RH, such as at $\lambda = 14$.

• *Page 7, Lines 121-123. This weak dissociation is actually responsible for another, different conductivity mechanism (i.e., likely a contribution to σ_{IP}).*

Response: In the paper mentioned by the reviewer (J. Am. Chem. Soc. 134, 19099-19107 (2012)), σIP is defined as: “*The electrical event ... is due to interfacial polarization (IP) associated with the conductivity σIP . The presence ... is typically observed in ionic conductors consisting of two or more phases ... results from the accumulation of charge at the interfaces between these phases.*” However, as we have explained in the previous section, our materials have randomly distributed hydrophilic side chain and hydrophobic polymer backbone. There are no phase separation, semi-crystalline structure, or ion clusters in the polymer, as evidenced by SAXS data in Supplementary Fig. 24. Therefore, while the σIP mechanism can explain interfaces between two phases, it does not account for the dissociation between cations and anions occurring at smaller length scales, particularly in a system without phase separation.

• Page 9, Lines 153-155. The authors must reconsider their interpretation also taking into account previous work discussing the vibrational spectra of hydrated ionomers, including: (see for instance: J. Phys. Chem. B, 113, 632-639 (2009); Solid State Ionics, 252, 84-92 (2013); Phys. Chem. Chem. Phys. 17, 4367-4378 (2015); and Phys. Chem. Chem. Phys. 21, 10357-10369 (2019)).

Response: We thank the reviewer for the comments. On Page 9, Lines 153-155, we indeed used FT-IR, a one-dimensional linear spectroscopy technique, to study water structure and cited representative work to support our interpretation. The papers by Dr. Vito Di Noto and his group, as mentioned by the reviewer, are also relevant. We have now cited these papers, which used FT-IR to understand water absorption and water structure, alongside other key references.

Changes to the manuscript: We have added some of the above references in the manuscript.

31. Hofmann, D. W. M. *et al.* Investigation of Water Structure in Nafion Membranes by Infrared Spectroscopy and Molecular Dynamics Simulation. *J. Phys. Chem. B* **113**, 632–639 (2009).

32. P. Pandey, T. *et al.* Interplay between water uptake, ion interactions, and conductivity in an e-beam grafted poly(ethylene-co-tetrafluoroethylene) anion exchange membrane. *Phys. Chem. Chem. Phys.* **17**, 4367–4378 (2015).

• Page 10, lines 173-177. The phenomena reported here are actually described elsewhere as “delocalization bodies” J. Am. Chem. Soc., 134, 19099-19107 (2012); and J. Am. Chem. Soc. 140, 1372-1384 (2018)).

Response: On page 10 lines 173-177, we aim to use graph theory to analyze the robustness of water percolation network and quantify the percolation network with edges. We don’t think this phenomenon can be quantified and described by delocalization bodies.

• Page 11, lines 195 and following. The authors must describe how the results of their simulation can be interpreted on the basis of the conductivity pathways σIP and σEP .

Response: We really appreciate that the reviewer pointed out these important concepts. However, we are interested in probing ion solvation, water dynamics, and water structures and understanding

the ion transport mechanisms at a molecular level. We have demonstrated above that the material systems where the ($\sigma_{EP} + \sigma_{IP}$) theory can be applied are very different from our material in many ways. We don't think the ($\sigma_{EP} + \sigma_{IP}$) theory will help us meet our research objectives.

• *Page 12, lines 208-208. The authors must describe how this statement fits with the two conductivity pathways σ_{IP} and σ_{EP} described elsewhere.*

Response: Please refer to the previous response.

• *Page 15, lines 261-271. This is simple speculation, without any experimental evidence. It is impossible that these processes, which occur at the 100-400 ps timescale, are the rate-determining steps of the conductivity which, as stated in Fig. 1, takes place with relaxation times longer than 1 microsecond.*

Response: This is not a simple speculation; rather, it is a well-established theoretical framework that has been validated in numerous studies over the decades. The reorientational dynamics of water, particularly the mechanisms and decoupling of reorientation, have long been a central focus in the study of water. The molecular jump mechanism of water reorientation, introduced in liquid water systems (Science 311, 832-835 (2006)), revolutionized our understanding of water dynamics and established the current view of hydrogen-bond reorientation. This mechanism has profound implications across fields such as chemistry and biology, influencing theories on solvation, hydrogen bonding, and protein folding (Annu. Rev. Phys. Chem. 62, 395-416 (2011)).

Subsequent studies extended this mechanism to define two initial states in various systems, including aqueous solutions ((PNAS 104, 11167-11172 (2007); J. Phys. Chem. B 112, 14230–14242 (2008)), salt solutions (Science 328, 1003-1005 (2010)), and ionic hydration shells (Science 328, 985-986 (2010); Science 328, 1006-1009 (2010)). Our study adopts this mechanism, examining water reorientation from these two initial states, defining different reorientation events, and calculating them using MD simulations—all of which align with established theory rather than speculation. All relevant references have been cited in the manuscript. Importantly, this is the first application of this mechanism in the context of ion transport in hydrated polymers.

Additionally, we did not state that any of these “are the rate-determining steps of the conductivity” on Page 15, lines 261-271. Our aim was to decouple the water reorientation and study its effects on ions at a molecular level.

• *Page 15, lines 277 and following. This statement is not clear.*

Response: We agree with the reviewer's comment and have revised the manuscript for improved clarity and precision.

Changes to the manuscript: We have modified the statement on page 15, line 277, and the following lines.

In low RH environments, there are very few water molecules in the Br^- second solvation shell, and they primarily form H-bonds with the water molecules that are in the first shell. These second-shell water molecules function as bridges (“bridge” waters, in the 2nd shell) that link adjacent first-

shell water molecules ("linked" waters, in the 1st shell), and consequently restrict their mobility. Notably, the "bridge" waters exhibit a consistent timescale for reorientation, as shown in Supplementary Fig. 37, and they govern the first regime observed in the anisotropy decay. Since Br⁻ ions are solvated by the "linked" waters in the 1st shell, their dynamics are slowed due to the low mobility of these "linked" waters, leading to a higher energy barrier that must be overcome for long-distance transport. This higher energy barrier corresponds to the calculated E_a in Regime I. Conversely, at higher RH levels, as the water population in the 2nd shell increases, the bridging effect is reduced, resulting in faster orientational dynamics for the 1st-shell waters. This allows Br⁻ ions to encounter a lower energy barrier, transitioning into the Regime II. This molecular-level description manifests the central role of the 3-edge ($k = 3$) in our k-stub theoretical analysis, connecting it to the observed dynamics.

• Page 16, lines 279-281. This is really confusing. The authors are comparing the activation energy E_a determined on the long-range migration processes with relaxation times longer than 1 microsecond with local motions with relaxation times on the order of picoseconds.

Response: Both of our IR and MD studies show that the anisotropy decay of water molecules that reflects the orientational dynamics of the water molecules falls into two regimes. From 25% RH to 55% RH, the material has a slower water orientational dynamics and above 55% RH the material has a faster water orientational dynamics. From EIS measurements, we observed the transition of ion transport happens at 55% RH. The Br⁻ transport is highly coupled with water structure and water dynamics. Although water reorientation motion occurs on the order of picoseconds, it is the collective effect of water dynamics and water structure that results in different ion transport, even though ion transport happens on a larger time scale. Additionally, as shown in Fig. 3c, ion transport involves displacements on nanometer scale that occur on a nanosecond timescale, bridging the gap between picosecond events and microsecond processes. It is believed by other researchers that the water reorientation on the order of picoseconds has a strong relationship with ion transport, as reported in *J. Phys. Chem. B* 2019, 123, 9408–9417 and *Phys. Rev. Lett.* **112**, 258301.

For better clarity and precision, we have revised the manuscript.

Changes to the manuscript: we have revised the manuscript on Page 16, lines 279-281 (see yellow highlighted section above).

• Page 16, lines 291 and following. These two interpretations are not new; they have already been extensively described in the literature also including results obtained by means of broadband electrical spectroscopy (BES). The results reported in this work do not detect any new event, but simply confirm findings widely accepted in the literature and interpreted on the basis of two (or more) conductivity pathways contributing to the overall conductivity of ionomers ($\sigma = \sigma_{IP} + \sigma_{EP}$)

Response: The polynorbornene homopolymer has randomly distributed hydrophilic side chain and hydrophobic polymer backbone. There are no phase separation, semi-crystalline structure, or ion clusters in the polymer. We are not sure if the theory is applicable to our material system. We really appreciate that the review pointed out these important concepts. However, we are interested in probing ion solvation, water dynamics, and water structures and understanding the ion transport mechanisms at a molecular level. We don't think the ($\sigma_{EP} + \sigma_{IP}$) theory will help us meet our

research objectives. We described the innovativeness of our work in the very first response and we believe the molecular-level information we provide in the manuscript for ion transport in AEM system has not been published elsewhere.

Reviewer #3

The manuscript presents a significant contribution to the field and is worthy of publication in Nature Communications, provided that the authors address the key issue regarding the justification for their conductivity predictions and highlight the limitation of the current methodology in providing the full picture for ion transport in AEMs. With this clarification, the paper will provide a robust and comprehensive understanding of water and ion dynamics, paving the way for future advancements in membrane technologies.

We thank the reviewer for the thorough and insightful comments.

1. The MD simulation study accounts for vehicular ion diffusion but does not consider the Grotthuss hopping mechanism, which is commonly known to play a significant role in ion conductivity. Despite this, the simulated conductivity matches the experimental values very closely, as provided in Figure 26 in the SI. The authors need to provide a robust justification for how their classical molecular dynamics (MD) calculations can accurately predict ion conductivity without incorporating the Grotthuss hopping mechanism. This is particularly important given the significant role that Grotthuss hopping is known to play in ionic conductivity (e.g., <https://doi.org/10.1021/acs.jpcc.8b02318>). In fact, I would like to know how do the authors reconcile the exclusion of Grotthuss hopping with the close quantitative match between the calculated and experimental conductivity values? Could there be a complementary mechanism or an overlooked aspect in the current simulation that compensates for the absence of Grotthuss hopping?

Response: We appreciate the reviewer's observation. Our study focuses on Br⁻ conducting AEMs which do not involve the Grotthuss mechanism due to the absence of OH⁻. This exclusion allows us to predict ion conductivity accurately using classical method, and match the experimental values closely. The accuracy of our MD models is further validated by a good agreement between the ultra-fast 2D IR data and MD data.

We have chosen to exclude the Grotthuss hopping in this study for the following reasons: (1) Excluding Grotthuss Hopping to Provide Insights into OH⁻ Conducting AEMs: Investigating Br⁻ conducting AEMs allows us to exclude the Grotthuss mechanism and focus solely on other potential mechanisms. Therefore, the close quantitative match between the calculated and experimental conductivity values reflects the accuracy of our simulation of bromide transport, not an oversight. This is particularly useful given the ongoing debate in previous research regarding the relative contributions of Grotthuss and vehicular mechanisms in OH⁻ conducting AEMs¹⁻² (J. Phys. Chem. Lett. 2018, 9, 825– 829, DOI: 10.1021/acs.jpcclett.8b00004, J. Am. Chem. Soc. 2016, 138, 991– 1000 DOI: 10.1021/jacs.5b11951). By focusing on bromide ion transport, we intentionally excluded the Grotthuss mechanism to thoroughly investigate the effects of water dynamics, water structure, and ion solvation in ion transport without the influence of proton hopping. We believe that a clear understanding of these effects will provide insights and inform our future computational studies on OH⁻ conducting AEMs. To study OH⁻ transport, advanced quantum mechanical methods, such as *ab initio* MD and QM/MM, and Machine-Learning Potentials can be employed.

(2) Applications of AEMs Conducting Halide Ions: We also emphasize the significance of studying Br⁻ conducting AEMs, even without OH⁻. AEMs that conduct halide ions have various applications, such as electrodialysis for desalination and brine treatment, chlor-alkali process in an electrolyzer, water treatment and purification, and bromine-based flow batteries. Understanding

the fundamental mechanisms of Br⁻ conducting AEMs can provide valuable insights into the molecular design of these membranes, with significant implications for these electrochemical applications.

2. Also, I believe the authors should highlight the potential limitations of the current simulation approach, and how might they affect the interpretation of the results?

Response: We thank the reviewer for the comment. While our current simulation accurately models AEMs conducting halide ions, such as Br⁻ in this study, it has limitations in capturing Grotthuss hopping events. To study OH⁻ transport, advanced quantum mechanical methods, such as *ab initio* MD and QM/MM, and Machine-Learning Potentials can be employed. Machine-Learning Potentials, in particular, show significant promise due to their speed and quantum-level accuracy in systems involving bond breaking and forming, enabling the capture of the Grotthuss mechanism. Our group has recently developed effective Machine-Learning Potentials, which will be employed in our studies of OH⁻ conducting AEMs in a separate paper.

By combining these advanced methods with classical MD simulations, we can achieve a comprehensive understanding of OH⁻ conducting AEMs.

To further address the concern, we have revised the abstract and discussion to clarify the scope of the work and highlight the potential limitations and their solutions.

Changes to the manuscript: We have added the discussion on the limitation of our simulation approach on page 17.

For study of OH⁻ transport, other computational methods, particularly quantum mechanical approaches such as *ab initio* MD and quantum mechanics/molecular mechanics (QM/MM), or alternative approaches based on machine learning (ML) potentials, are necessary to capture proton hopping mechanisms. By combining these quantum mechanical and ML methods with classical MD simulations, we hope to achieve in future work a comprehensive understanding of OH⁻ conducting AEMs.

3. One very minor point is that in Figure 3h in the manuscript, if the y-axis shows the probability density, the unit should be ns⁻¹.

Response: We thank the reviewer for pointing this out. We have updated Figure 3h to reflect the correct unit of ns⁻¹ on the y-axis.

Changes to the manuscript: We changed the unit of y-axis in Fig.3h.

Reviewer #4

This paper carefully verifies and supports this concept using 2D-IR and MD. Although the results obtained are rather within the range of expectation, it is a very interesting paper in that it is clarified by careful MD simulations and experiments. On the other hand, although a comment is made in the conclusion about the ideal molecular structure of AEM that realizes fast ionic conductivity, it cannot be said that the results are particularly new. In this paper, water is studied and there is no discussion of the influence of the polymer or how the type of ion or the strength of the interaction affects the mechanism. Although the polymer certainly moves slowly and can be omitted from the discussion, it affects the dispersion of the water molecules and must be considered in the molecular design of the AEM. In conclusion, however, the paper is based on thorough experiments and in-depth considerations based on comparisons with MD, and may be considered for publication in Nature Communications.

Response: We thank the reviewer for the thorough and constructive comments. In our study, we introduce new methodologies for understanding the fundamental relationship between water structure, water dynamics, and ion transport in polyelectrolytes. Specifically, we demonstrate for the first time that a percolation network requires at least three edges to facilitate faster ion transport, a finding that had not been quantified before. Additionally, our simulations successfully replicate the experimental 2D IR results, showing that an increase in the population of water molecules in the second solvation shell is necessary to eliminate the bridging effect and enable fast ion transport. This is the first application of 2D IR in the context of ion transport in hydrated polymers, making both of these findings novel contributions to the field. We have also addressed the reviewer's concerns regarding the dispersion of water and polymer configurations in the following sections.

1. MD results depend on the initial configuration. Especially in the low RH region, the dispersion of water molecules may not be reproducible. Since it seems that only one model is considered in this paper, it is desirable to study and compare results for different initial configuration to strengthen the validity in the low RH region.

Response: We thank the reviewer for the comments, and agree with the suggestion. To validate our findings, particularly in the low RH region, we conducted three additional independent MD simulations from scratch. For each RH level ranging from 25% to 45%, we prepared three samples using different initial configurations. The polymer chains and ions were randomly placed in a large simulation box with unique random seeds to ensure both uniqueness and reproducibility. Water molecules were then added to each system to achieve the corresponding water content. The equilibrium and production procedures followed the same protocol as described in the SI.

In Supplementary Fig. 38, we present snapshots of the simulation box without water. It shows that each independent simulation starts from a distinct initial configuration. After randomly adding water, the densities of each system at the same RH are within the RMSD. The averaged values along with error bars are provided in Supplementary Information, Section 3.1, Table 6.

In the supplementary information, we have included several key analyses for each RH level and each independent simulation. These analyses confirm that the conclusions remain consistent

regardless of the initial configurations. Since polymers in the low RH region are more immobile and could have a larger effect on the dispersion of water molecules, the fact that the additional analyses do not change the conclusions in the low RH region suggests that the results would also remain unchanged in the high RH region. We think this consistency is attributed to the randomly distributed hydrophilic functional groups and hydrophobic polymer backbone, which is evidenced in the SAXS data shown in Supplementary Fig. 24.

We have added the above discussion to the supplementary information.

Supplementary Fig. 38 | Snapshots of three additional independent simulations starting from different initial polymer configurations.

Changes to the manuscript: Supplementary Information, page 56-57, Supplementary Section 5.

2. *Information on the number of water molecules in the model is needed. Also, information on density or cell length would be useful to researchers performing similar calculations.*

Response: We thank the reviewer for the comments. We have included the hydration number and equilibrium density for different systems in the Supplementary Information, **Section 3.1, Table 6**. To enhance reproducibility, we provided the hydration number rather than the number of water molecules. The number of ions is also detailed in Section S3.1, so the number of water molecules can be obtained by multiplying the hydration number by the number of ions. We have also added more details for better clarification, which are highlighted on **page 37-39** of the SI.

Changes to the manuscript: Supplementary Information, page 37-39, Supplementary Table 6.

3. *Fig.2. c: It is necessary to mention that the RH dependence of CN is different from a and b.*

Response: On **page 7, para. 1** of the manuscript, we have already compared the RH dependence of CN for Br⁻-N (Fig. 2c) with Br⁻-Ow (Fig. 2b). Specifically, we stated: “As RH increases, a

transition emerges: the coordination of the ammonium groups diminishes as the number of water molecules increases, a trend that can be visualized in Fig. 2d-f. This suggests that water molecules play an important role in the solvation of anions by reducing their interaction with ammonium groups.” This also implies that the increase in water molecules should result in an increasing trend of CN for Ow-Ow in Fig. 2a, similar to Fig. 2b, and opposite to Fig. 2c. We have now made this distinction more explicit in the manuscript to address the concern.

4. Fig.3 h: Although the distribution of time to cross a distance of 1 nm is shown, there are likely many molecules that frequently enter and exit the boundary. The effect of these may be large and change depending on the sampling interval.

Response: We appreciate the reviewer’s concern and concur with that. To address this, we have calculated the FPT distribution using different sampling intervals ranging from 10 ps to 500 ps, which provides a wide range for ion transport events. Supplementary Fig. 39a-b show that the distribution curves overlap and display no significant changes when changing sampling intervals. This also indicates that, for both low (Supplementary Fig. 39a) and high RH (Supplementary Fig. 39b), the choice of sampling interval does not affect the FPT distribution trend in our calculations.

Supplementary Fig. 39 | First Passage Time (FPT) distribution for ion mobility at different sampling intervals. FPT distributions are calculated with sampling intervals ranging from $dt = 10$ ps to 100 ps at **a**, 25% and **b**, 85% RH. The curves for different sampling intervals overlap closely at both low and high hydration level, indicating that the sampling interval has minimal effect on the FPT distribution. The sampling interval of 20 ps is the original interval used to calculate the FPT distribution in Fig. 3h.

The minimal effect observed in the FPT distribution across different sampling intervals can be attributed to the dominance of long-range ion transport dynamics over short-range, transient boundary crossings. This confirms that the choice of 1 nm as the distance for measuring FPT is sensible, as it represents a long-range transport event compared to boundary crossings. Ions frequently entering and exiting the boundary undergo rapid, local movements that do not significantly impact the overall FPT. These local movements are averaged out in the measurement, which focuses on the time required for an ion to traverse 1 nm. The consistency in the FPT distribution across various sampling intervals ensures the robustness of our results. Additionally, it holds for both low (25% RH) and high (85% RH) hydration levels, further confirm that this

effect is negligible. For better clarification, we have added the above discussion to the revised Supplementary Information, **page 57-58**.

Changes to the manuscript: Supplementary Information, **page 57-58**, Supplementary **Section 5.1, Fig. 39**.

5. Fig.3: Is the k related to the hydration number? If so, please discuss it together with CN in Fig.2.

Response: Although the k-stub analysis indicates the transition in percolation, k is not directly related to the hydration number (λ). The parameter k indicates the number of hydrogen bonds required for a robust cluster, and the potential bottleneck. While λ is a macroscopic quantity that affects the coordination number (CN), the k-stub analysis focuses on the local solvation structure at the molecular level. To be specific, "edges" are defined as identified hydrogen bonds. Therefore, the same λ does not necessarily result in the same topology of hydrogen bond networks. In fact, in our forthcoming paper, we found that for different polymer structure systems, percolation transition occurs at different λ values even for the same k value.

Response to Reviewers' Comments

Manuscript ID: NCOMMS-24-35031-A

Manuscript Title: Water Dynamics, Water Structure, and Ion Transport in an Anion Exchange Membrane

We would like to thank the reviewers for taking the time to provide their thorough and insightful comments. We have carefully considered each comment and have revised the manuscript accordingly. Our point-by-point response to specific comments and the changes made are detailed below. Page and paragraphs (**para.**) numbers quoted below refer to those in the marked manuscript file (with yellow highlighter).

Reviewer #2

Though the authors have addressed several of the concerns raised by the reviewer, there are still several fundamental points that were not considered properly, pointing to a severe misunderstanding of the phenomena associated to the ion transport phenomena.

Response: We disagree with the assertion that we misunderstand the ion transport phenomena in our polymer. On the contrary, our study provides pivotal insights by correlating water structure and water dynamics, occurring on the scale of tens of picoseconds, with ion transport over nanometer length scales. We believe this is the first time such a correlation has been established in a polymer electrolytes system, offering a deeper understanding of the molecular mechanisms governing ion transport. We have thoroughly addressed the points raised and have made revisions to the manuscript to ensure clarity.

1. The ion-transport mechanism involving different conductivity pathways can also be applied to homogeneous systems that do not exhibit the clear phase separation of hydrated perfluorinated ionomers such as hydrated Nafion. The ion-transport mechanism involving different conductivity pathways has also been applied to homogeneous systems such as ionic liquids (see for instance: J. Power Sources 565, 232910 (2023)). The different domains along whose boundaries the charge can accumulate need not correspond to different hydrophilic/hydrophobic domains. They may simply correspond to different domains (e.g., areas where the packing of the macromolecules is only slightly different) within an otherwise homogeneous system (e.g., the polynorbornene homopolymer considered by the authors). This point was not addressed at all in the revised version of the manuscript, and incorrectly leads the authors to not apply the well-established s(EP+IP) model to their system.

Response: We acknowledge that the theory of multiple conductivity pathways can be applied to systems like Nafion ionomers. However, we would like to clarify that Nafion is widely recognized as a semi-crystalline and nanostructured material exhibiting phase-segregation morphology, rather than being homogeneous. This is evidenced by ionomer peaks observed in SAXS/WAXS measurements (see, for example, the well-recognized review in Chem. Rev. **117**, 987–1104 (2017), DOI: [10.1021/acs.chemrev.6b00159](https://doi.org/10.1021/acs.chemrev.6b00159)).

Regarding the application of the multiple conductivity pathways model to other homogeneous systems, the reviewer refers to a study on pyrrolidinium-magnesium-organochlorostannate ionic liquid electrolytes for metal batteries (J. Power Sources **565**, 232910 (2023)). We note that this multivalent metal-based ionic liquid material system is fundamentally different from our polynorbornene homopolymer. The cited material system contains grain boundaries that affect ion transport. The material exhibits a crystallization transition temperature (as shown in Fig. 1 of the referenced paper), indicating a crystalline or semi-crystalline nature. Such structural features can significantly influence ion transport mechanisms.

In contrast, our system is based on a polynorbornene homopolymer, which does not have crystalline domains or grain. Therefore, the multiple conductivity pathways model, which accounts for transport along different domains or interfaces, may not be directly applicable to our system.

Furthermore, we acknowledge the importance of techniques like Quasi-Elastic Neutron Scattering (QENS) and Broadband Electrical Spectroscopy (BES) in studying ion transport phenomena, as mentioned in the introduction of our manuscript. However, our approach focuses on investigating water structure and dynamics at the molecular level—from femtoseconds to picoseconds to microseconds—within state-of-the-art anion exchange membranes (AEMs). Our model uncovers the ion transport mechanism at the atomistic level, which is fundamentally different from the macroscopic model referenced by the reviewer. By correlating these water dynamics with ion transport over nanometer length scales, we aim to provide new insights into ion transport mechanisms that are not accessible through the multiple conductivity pathways model.

We have revised the manuscript to further clarify these points and to address the reviewer's concerns.

Changes to the manuscript:

Here we note that quasi-elastic neutron scattering (QENS)²³ has previously been used to examine water dynamics at picosecond timescales, and broadband electrical spectroscopy (BES) has been employed to investigate ion conducting pathways through grain boundaries in polyelectrolytes^{24–26}.

2. If the temperature range is small enough, the ion conductivity any system can be approximated with an Arrhenius trend. At the very least this point should have been mentioned clearly in the revised text, including the corresponding justification for such a gross approximation. In addition, the absence of a clear Tg at T > 0°C (the range measured by the authors) may simply indicate that the Tg actually occurred below 0°C. Thus, there is no proof that prevents the proposed materials to follow a VTF conductivity trend, leading to incorrect analysis and conclusions.

Response: Our goal is to understand water structure, water dynamics, and ion transport above 0 °C to guide the design of polyelectrolytes for electrochemical systems, including fuel cells, water electrolyzers, CO₂ electrolyzers, redox flow batteries, and reverse electrodialysis, which typically operate at or above room temperature. We believe that investigating ion transport below 0 °C is unlikely to contribute to the advancement for these technologies.

Below, we clearly show that **within the entire operational temperature range, the polymer remains in a glassy state**. Therefore, the use of the Arrhenius model is justified, while applying the Vogel-Tammann-Fulcher (VTF) model would be fundamentally incorrect.

To determine the glass transition temperature (T_g) of our polynorbornene polymer, we conducted comprehensive Differential Scanning Calorimetry (DSC) experiments:

1. First DSC Set (Supplementary Fig. S20): We measured dry polynorbornene up to 250 °C and did not detect any T_g . This indicates that the polymer does not exhibit a glass transition within this temperature range.
2. Second DSC Set (Supplementary Fig. S21): We measured hydrated polynorbornene from –60 °C to 90 °C. Again, no T_g was detected, even in the fully hydrated state. This suggests that the polymer remains glassy throughout our experimental conditions.

To further validate these findings, we calculated Bond Vector Autocorrelation Function (BVAf) in simulations to assess the polymer dynamics. The BVAf results (as shown in Supplementary Fig. S27) reveal that the polymer backbone, side chains, and functional groups exhibit extremely slow dynamics at room temperature, consistent with behavior below T_g . This supports our conclusion that the polymer is in a glassy state during all ion transport and 2D IR measurements.

Moreover, **vinyl-addition polynorbornene is well-documented for its high T_g** , typically around **300 °C or higher**, as reported in several studies:

1. *Polymer* 2020, 203, 122759. <https://doi.org/10.1016/j.polymer.2020.122759>;

2. *J. Polym. Sci. B Polym. Phys.* 1999, 37, 3003– 3010.

[https://doi.org/10.1002/\(SICI\)1099-0488\(19991101\)37:21<3003::AID-POLB10>3.0.CO;2-T](https://doi.org/10.1002/(SICI)1099-0488(19991101)37:21<3003::AID-POLB10>3.0.CO;2-T);

3. *Macromolecules* 2002, 35, 8933. <https://doi.org/10.1021/ma025586j>

Given this evidence, we are confident that our polymer remains in the glassy state across the temperature range studied. **The application of the Arrhenius model is appropriate under these conditions**, as the ion transport is not coupled with polymer segmental motions, which are negligible below T_g . The VTF model is typically applicable to polymers above T_g , where temperature-dependent segmental dynamics significantly influence ion conductivity.

We have updated the manuscript to explicitly justify our choice of the Arrhenius model.

Changes to the manuscript:

It is important to note that PBBNB⁺Br⁻ remains in a glass state throughout all EIS measurements, as no glass transition temperature (T_g) was detected in the differential scanning calorimetry (DSC) measurements (Supplementary Fig. S20 and S21). Therefore, the plot of $\ln \sigma$ versus $1000/T$ follows Arrhenius behavior for each RH, allowing us to fit the data and determine the activation energy (E_a).

3. Finally, the authors have not considered appropriately the issue of different timescales (water vs. the entire system) owing to their reluctance to consider appropriately the issue of multiple conductivity pathways in an apparently homogeneous system.

Response: We have addressed the issue of different timescales in the previous revised version of the manuscript, but we would like to clarify further.

In our system, the dynamical behavior of water occurs on the scale of tens of picoseconds and exhibits different behaviors in low and high relative humidity (RH) regimes. This leads to variations in water mobility within the first and second solvation shells, affecting ion mobility due to changes in the ion solvation environment. Notably, this timescale aligns with that of local ion fluctuations. As time progresses from tens of picoseconds to nanoseconds, ions travel over nanometer length scales, as reported in Fig. 3h-i. In other words, ion transport events (nanometer movement over nanoseconds) are the cumulative result of these smaller timescale events (tens of picoseconds involving changes in local solvation environments).

Regarding polymer dynamics, we have shown in Supplementary Fig. 27, through bond vector autocorrelation function (BVAf) analyses, that all polymer dynamics – whether in the backbone, side chains, or functional groups – are extremely slow. The effects of polymer dynamics are negligible within the tens of nanoseconds timescale, allowing us to isolate the water and ion dynamics without interference from polymer motion.

Therefore, through multiple analyses, we have connected events occurring at different timescales – from picoseconds to nanoseconds – considering water, ions and the entire system. There is no issue regarding different timescales.

Concerning the *multiple conductivity pathways* mentioned by the reviewer, we have carefully reviewed the reference literature and found that this method does not access the picosecond dynamics of water molecules. While multiple conductivity pathways may explain certain phenomena at a macroscopic level, they are less effective for investigating water dynamics at the molecular level. However, given the presence of different timescales in our system, as the reviewer acknowledged, we believe that incorporating more explicit techniques is essential for understanding ion transport mechanisms and their interrelation with water. In this regard, our method contributes to the field by providing molecular-level insights that benefit the broader scientific community.

A detailed description of this mechanism is provided in the manuscript, and we have revised discussions for better clarification.

Changes to the manuscript:

By decoupling these modes, we find that water behaving as 'W-W Jump' and 'W-W Frame' exhibits this bi-modal characteristic within our observational timeframe; additional details are presented in Supplementary Fig. 37. In low RH environments, there are very few water molecules in the Br⁻ second solvation shell, and they primarily form H-bonds with the water molecules that are in the first shell. These second-shell water molecules function as bridges (“bridge” waters, in the 2nd shell) that link adjacent first-shell water molecules (“linked” waters, in the 1st shell),

consequently restricting their mobility. Notably, the "bridge" waters exhibit a consistent reorientation timescale on the order of tens of picoseconds, as shown in Supplementary Fig. 37, and they govern the first regime observed in the anisotropy decay shown in Fig. 4e-f. This timescale aligns with the local fluctuations of Br^- ions shown in Fig. 3i. As time progresses from tens of picoseconds to nanoseconds, these local interactions accumulate, allowing Br^- ions to traverse nanometer-length scales, as displayed in Fig. 3h-i. In other words, ion transport over nanometers in nanoseconds is the cumulative result of these shorter timescale events involving changes in the local solvation environment.

Since Br^- ions are solvated by the "linked" waters in the first shell, their dynamics are slowed due to the low mobility of these "linked" waters, leading to a higher energy barrier that must be overcome for long-distance transport. This higher energy barrier corresponds to the calculated E_a in Regime I. Conversely, at higher RH levels, as the water population in the second shell increases, the bridging effect is reduced, resulting in faster orientational dynamics for the first-shell waters. This accelerates local ion fluctuations and reduces the energy barrier for ion transport, transitioning into Regime II. This molecular-level description demonstrates how events at different timescales are interconnected, and manifests the central role of the 3-edge ($k = 3$) in our k -stub theoretical analysis, connecting it to the observed dynamics.

Response to Reviewers' Comments

Manuscript ID: NCOMMS-24-35031-B

Manuscript Title: Water Dynamics, Water Structure, and Ion Transport in an Anion Exchange Membrane

We would like to thank the reviewers for taking the time to provide their thorough and insightful comments. We have carefully considered each comment and have revised the manuscript accordingly. Our point-by-point response to specific comments and the changes made are detailed below. Page and paragraphs (**para.**) numbers quoted below refer to those in the marked manuscript file (with yellow highlighter).

Reviewer #3

1. The authors correctly noted that the T_g of this polymer is above 300 °C (based on the references provided from the literature); however, their reference to the extremely slow dynamics of the polymer and side chain in simulations is not entirely accurate. Although such slow dynamics compared to those of the water and ions support their argument—and I believe this should be noted in the paper—it does not prove that the actual T_g of the polymer is above the simulation temperature. In fact, due to the extremely smaller time frame of simulations compared to experimental tests (such as DSC), even soft polymers can exhibit very limited segmental motions in atomistic simulations.

Response: We appreciate the reviewer's feedback and agree that the extremely small time frame of simulations compared to experimental tests (e.g., DSC) can influence the observed polymer dynamics. However, even within the limited simulation time scales, soft polymers and glassy polymers exhibit distinct dynamics in bond vector autocorrelation function (BVAf) measurements, as shown in prior work (Macromolecules 2020, 53, 8, 2783–2792, Figure 4a). Notably, even for polymers below their glass transition temperature (T_g) at the simulated temperature, BVAf results indicate that our polymer exhibits slower dynamics compared to those reported in the referenced work. This suggests that our material is more rigid and exhibits very limited segmental motion.

To address the reviewer's concern, we have revised the discussion to focus less on the direct relationship with T_g and more on the relative dynamics within our system. Specifically, we emphasize that the polymer dynamics are significantly slower compared to those of water and ions. This is supported by our mean square displacement (MSD) calculations, which directly compare the mobility of polymer atoms, bromide ions, and water molecules. At both low and high hydration levels (25% and 85% RH), the MSD of polymer atoms is 3–4 orders of magnitude lower than that of ions and water, as shown in Supplementary Fig. 27d-e. This trend is consistent across all hydration levels, confirming that the polymer dynamics are negligible relative to the dynamics of water and ions in our system.

We believe this revised framing provides a more rigorous and straightforward interpretation of our results, aligning with the reviewer's suggestion.

Changes to the manuscript: We have changed the words and added a discussion on the polymer dynamics analysis on **page 42-43** of the Supplementary Information.

In our study, we have defined two types of vectors, which capture relaxation phenomena at the polymer backbone level and at the sidechain reorientation level. Supplementary Figs. 27b-c demonstrate that both types of vectors, across all systems, exhibit slow relaxation dynamics. This observation confirms that all our systems exhibit slow polymer dynamics at room temperature.

To compare the polymer dynamics to those of the water and ions, we calculated the MSD of polymer atoms, bromide ions, and water molecules across all RH levels. Supplementary Figs. 27d-e demonstrate that, at both low and high hydration levels, the MSD of polymers is three to four orders of magnitude lower than that of the ions and water. This indicates that, across the entire range of RH investigated in our work, polymer dynamics are sufficiently slow to be considered negligible compared to the dynamics of ions and water.

2. To reconcile the true concern of the reviewer with the response of the authors, I think the authors could soften the tone about the homogeneity of the polymer while arguing that although it is an assumption, it is a valid and robust one, as there is no evidence of crystallinity, phase separation, or ordered/disordered domains reported or found for this membrane.

Response: We agree with the reviewer for the comment. We have softened our tone about the homogeneity of the polymer in the manuscript.

Changes to the manuscript: We have made the following changes in the manuscript on **page 6**:

In addition, we assume our polymer system to be homogeneous, as there is no evidence of crystallinity, phase separation, or ordered/disordered domains (Supplementary Fig. 24).

3. Additionally, the negligible dynamics of polymer compared to water and ion, as evidenced from both DSC measurements and MD simulations (I would suggest the inclusion of mean squared displacement curves for the atoms of the polymer backbone and sidechain and water molecule in one graph to be comparable), provides extra support for the justification of the conclusions of the paper.

Response: We thank the reviewer for this suggestion. In response, we have calculated and plotted the mean square displacement (MSD) curves for the atoms of the polymers, ions, and water molecules across RH levels. For better comparison, we have combined these results into a single graph for the lowest and highest RH levels, as shown in Supplementary Fig. 27d-e.

The manuscript has been revised to include this analysis and discussion, highlighting how these combined MSD curves provide additional support for the conclusions regarding the negligible dynamics of the polymer compared to water and ions.

Changes to the manuscript: Supplementary Information, **page 42-43**.

Supplementary Fig. 27 | Polymer dynamics characterized using the bond vector autocorrelation function (BVAf) and mean square displacement (MSD). a, Schematic illustrating two types of bond vectors used in calculations. b-c, BVAf for the (b) backbone bond and (c) sidechain-terminal vector. d-e, MSD of polymer atoms, bromide ions, and water molecules at (d) 25% RH and (e) 85% RH.